# DC-ESN: Diffusion Convolutional Echo State Network for Spatiotemporal Traffic Forecasting

## Abstract

Traffic forecasting is a challenging spatiotemporal problem that requires capturing complex dependencies across both road networks and time. Existing deep learning approaches such as Convolutional Neural Networks (CNNs) for spatial modeling, Recurrent Neural Networks (RNNs), Long Short-Term Memory (LSTM), and Gated Recurrent Unit (GRU) networks for temporal modeling, as well as Graph Convolutional Networks (GCNs) for capturing topological structures have achieved notable success. But they often incur high computational cost and require large number of trainable parameters. In this work, we introduce a novel architecture named Diffusion-Convolutional Echo State Network (DC-ESN) designed for spatio-temporal forecasting, which combines diffusion convolution for spatial feature extraction with an Echo State Network (ESN) for efficient temporal modeling. This structural decoupling allows the model to learn complex spatial topologies via gradient descent while leveraging the asymptotic stability of the fixed reservoir for temporal memory, offering a robust and computationally efficient alternative to fully trainable spatiotemporal graph networks. Compared with Diffusion-Convolutional Recurrent Neural Networks (DCRNN), the proposed model DC-ESN achieves comparable predictive accuracy while significantly improving inference efficiency and reducing GPU memory usage. Experiments on the METR-LA and PEMS-BAY benchmark traffic datasets demonstrate that DC-ESN attains faster inference with minimal accuracy loss, making it suitable for real-time forecasting applications.

**Keywords:** Diffusion convolution, Echo state network, Graph neural network, Reservoir computing, Spatiotemporal forecasting, Traffic forecasting

## 1 Introduction

Spatiotemporal forecasting is a crucial task in many scientific and engineering domains, including weather modeling, environmental monitoring, and traffic prediction (Kumar et al., 2024). It aims to predict future observations that depend on both spatial and temporal relationships (Griffith, 2010). Applications of spatiotemporal forecasting include traffic flow prediction (Ermagun & Levinson, 2018), energy demand forecasting (Lin et al., 2018) and air quality estimation (Tascikaraoglu & Sanandaji, 2016). The main focus of this research is traffic forecasting on road networks, which constitutes a fundamental component of Intelligent Transportation Systems (ITS) (Bazzan & Klügl, 2022). The purpose of traffic forecasting in this work is to anticipate future traffic speeds for a sensor network based on past traffic speeds and the underlying road network structure (Chan et al., 2012). This is a challenging task due to the inherent nonlinear nature of traffic flow, complex spatial correlations across road networks, and strong temporal dependencies over time (Tedjopurnomo et al., 2020) (Carianni & Gemma, 2025).

Traditional machine learning methods such as Support Vector Regression (SVR) (Gong et al., 2013), Random Forests (RF) (Zarei et al., 2013), and Autoregressive Integrated Moving Average (ARIMA) (Kumar & Vanajakshi, 2015) have been widely applied to this problem, but their ability to capture nonlinear and high-dimensional spatiotemporal relationships is limited. Traditional time-series models such as ARIMA

and Kalman filters assume linear and stationary processes, that restricts their ability to model nonlinear dependencies (Lippi et al., 2013). Traffic time series exhibit significant temporal dynamics, and non-stationarity caused by recurring events, such as accidents or rush hours, can make long-term forecasting challenging (Cheng et al., 2021).

In many classical deep-learning tasks (e.g., image recognition), data lie on a regular Euclidean grid; pixels are arranged in rows and columns, and convolution operations assume uniform neighbourhoods (Lecun et al., 1998; Hinton et al., 2012). Traffic networks, however, do not conform to this structure. As noted in the literature (Shuman et al., 2013; Xing et al., 2023); because transportation networks are non-Euclidean in structure, dividing the network into grids for convolutional operations using CNNs destroys important structural information. This drawback is illustrated in Figure 1. Recent advancements in deep learning, such as Graph Neural Networks (GNNs), have improved performance in non-Euclidean domains.

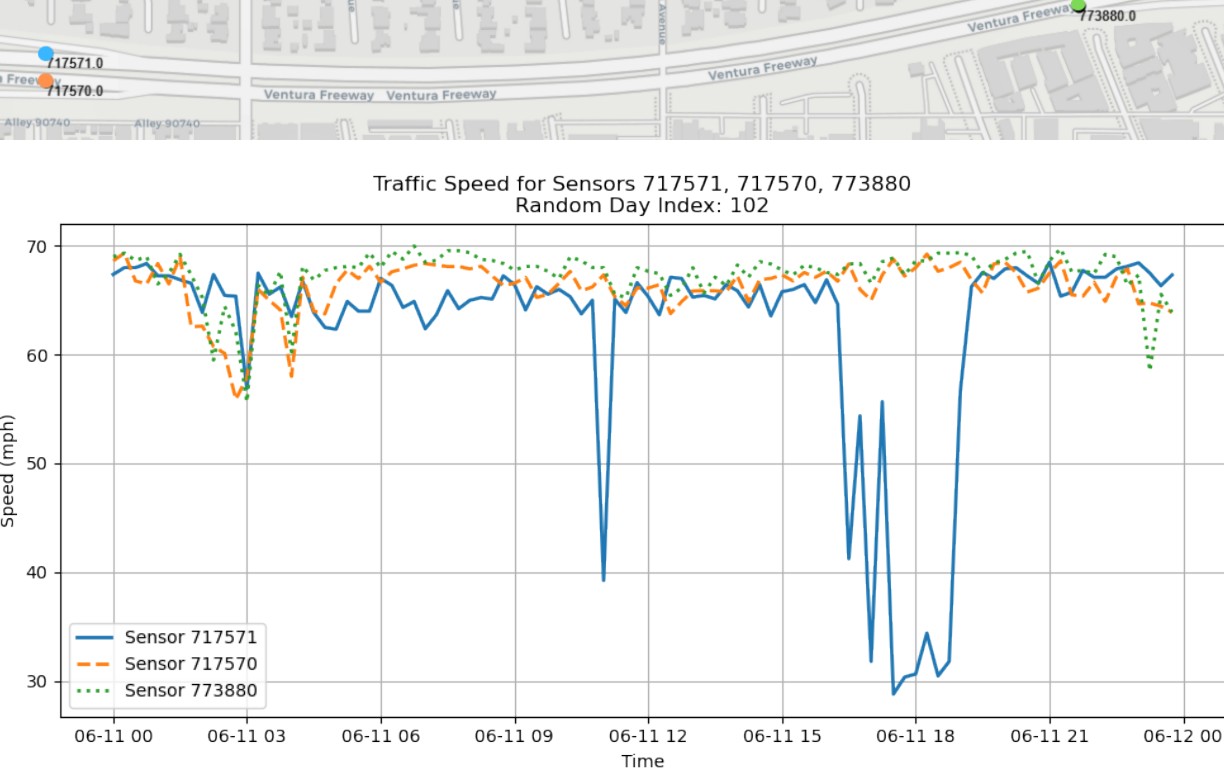

Figure 1: Speeds at sensors 717570 and 773880 are correlated because they are located on the same road, whereas the speed at sensor 717571 follows a different pattern, indicating no spatial relation between sensors that are close to each other in the Euclidean domain (sensor 717571 and 717570).

The Diffusion Convolutional Recurrent Neural Network (DCRNN) (Li et al., 2018) is a graph-based model that introduces the idea of modeling spatial dependencies using diffusion processes on directed graphs, combined with Gated Recurrent Units (GRUs) for temporal dynamics. Although GRUs and Long Short-Term Memory (LSTM) networks have demonstrated strong capabilities in capturing long-range temporal dependencies, their training involves backpropagation through time (BPTT), which is computationally expensive and prone to vanishing or exploding gradients (Bengio et al., 1994; Hochreiter & Schmidhuber, 1997; Pascanu et al., 2013).

To address these challenges, we replace GRUs with a lightweight and efficient reservoir computing model known as the Echo State Network (ESN) (Jaeger, 2001). ESNs consist of a large, fixed recurrent reservoir with randomly initialized and untrained internal weights, while only the output (readout) layer is trained. This structure allows ESNs to retain the temporal representational power of recurrent models while significantly reducing computational cost and training time.

Subsequent developments in Echo State Networks (ESNs) (Gallicchio et al., 2018; Gallicchio & Micheli, 2017), further extend ESNs by stacking multiple reservoirs hierarchically to capture multi-scale temporal dependencies. Leveraging these properties, we propose the Diffusion Convolutional Echo State Network (DC-ESN), which integrates diffusion convolution for spatial feature extraction with ESNs for efficient and scalable temporal modeling. This hybrid design preserves DCRNN's ability to model spatiotemporal correlations while achieving faster training and inference due to the non-trainable recurrent dynamics of the ESN. The proposed architecture achieves up to 5.5x faster training time and 64% lower GPU usage than DCRNN, with only 3-6% higher MAE, making it suitable for real-time deployment.

## 2  Related Work

Traffic forecasting has been an active research topic in both transportation engineering and machine learning, focusing on modeling complex spatiotemporal dependencies in road  (Yin et al., 2021). In general, existing methods can be categorized into three main groups: Statistical approaches, Classical machine learning approaches, and Deep learning approaches. Each category contributes unique modeling perspectives but also faces inherent limitations when dealing with highly nonlinear and complex spatiotemporal dependencies.

### 2.1  Classical Time-Series Forecasting Methods

Early approaches were dominated by statistical and classical machine learning methods such as Autoregressive Integrated Moving Average (ARIMA) (Kumar & Vanajakshi, 2015), Vector Autoregression (VAR), Kalman filtering, Support Vector Regression (SVR) (Gong et al., 2013), Random Forests (RF) (Zarei et al., 2013), and k-Nearest Neighbors (kNN). Although these techniques achieved moderate success, they relied heavily on handcrafted features and were limited in handling nonlinear relationships and large-scale, high-dimensional data.

### 2.2  Spatiotemporal Deep Learning Models

With the advent of deep learning, models such as Recurrent Neural Networks (RNNs), Long Short-Term Memory (LSTM), and Gated Recurrent Units (GRU) (Wang et al., 2020) have been employed to capture temporal dependencies, while Convolutional Neural Networks (CNNs) have been utilized to model spatial correlations by treating traffic data as grid-like images. However, CNN-based approaches struggle with non-Euclidean road network structures (Li et al., 2018) . To address this, Graph Neural Networks (GNNs) have emerged as powerful tools for modeling graph-structured data. In particular, Graph Convolutional Networks (GCNs) (Kipf & Welling, 2016) and Chebyshev Networks (ChebNet) (Defferrard et al., 2016) extended CNN operations to irregular graph domains, while Diffusion Convolutional Recurrent Neural Networks (DCRNN) (Li et al., 2018) modeled traffic flow as a diffusion process on directed graphs to capture asymmetric spatial dependencies.

To achieve precise traffic signal forecasting, it is essential to first address spatial modeling, a foundational requirement for analyzing spatiotemporal traffic data. Current methodologies for spatial modeling in this field can be broadly organized into four primary paradigms:

1. **Identity Matrix Representations:** Treating the problem as standard multivariate time-series forecasting without explicit spatial links.

2. **Pre-defined Adjacency Matrices:** Utilizing static, domain-driven graph topologies to encode fixed spatial relationships (Shuman et al., 2013; Li et al., 2018).

3. **Trainable Adjacency Matrices:** Learning a static latent graph structure via node embeddings to discover hidden dependencies (Wu et al., 2019b).

4. **Attention-based Mechanisms:** Employing dynamic spatial modeling, such as Transformers or Graph Attention Networks, to capture time-varying dependencies without prior structural knowledge (Park et al., 2020; Zheng et al., 2020).

## 2.3 Graph-Based Traffic Forecasting

Graph WaveNet (Wu et al., 2019a) represented a significant shift in spatial modeling by introducing node embeddings to generate learnable adjacency matrices, thereby overcoming the limitations of fixed, distance-based graphs. Building on the foundations laid by Graph WaveNet and DCRNN (Li et al., 2018), subsequent research has explored various learnable graph architectures, including AGCRN (Bai et al., 2020) and MTGNN (Wu et al., 2020). While learnable static graphs have notably advanced the state-of-the-art, contemporary research has further extended these capabilities by modeling time-varying graph structures that adapt to evolving network dynamics.

## 2.4 Transformers and Attention-based Models

Recent advancements have shifted toward capturing more dynamic spatiotemporal dependencies using attention mechanisms and Transformers like STPT (Kumar et al., 2024), GMAN (Zheng et al., 2020), Informer (Zhou et al., 2021), Autoformer (Wu et al., 2021), and STAEformer (Liu et al., 2023) etc. ST-GRAT (Park et al., 2020) and other attention-based models utilize multi-head self-attention to model dynamic spatial correlations without being restricted by pre-defined graph topologies (Zheng et al., 2020) (Wu et al., 2019b). While Transformers excel at capturing long-term dependencies in parallel, their quadratic complexity in both time and memory remains a primary limitation compared to traditional architectures. These models often consume considerable memory resources and suffer from bottleneck problems caused by autoregressive decoding (Park et al., 2020) (Lee & Ko, 2024). Time-varying or dynamic graph modeling is noise sensitive. Recent studies also report that they may fail to generate informative attention maps by spreading weights too uniformly across the network (Jiang et al., 2023). Newer models specifically designed to address the "computational efficiency tradeoff" are being developed which include Fedformer (Zhou et al., 2022) EGFormer (Yang et al., 2024)

Furthermore, to address the limitations of pre-defined graphs that ignore dynamic traffic changes, memory networks based models have been developed. Pattern Matching Memory Networks (PM-MemNet) have been proposed to transform forecasting into a key-value pair matching task, allowing models to "remember" long-term historical patterns (Lee et al., 2021). MegaCRN integrates memory networks with meta-graph learning to construct noise-robust, time-varying graph structures (Jiang et al., 2023). Additionally, recent studies suggest that strategic expert selection can outperform massive ensemble complexity, highlighting a shift toward more nuanced modeling strategies (Guettala et al., 2025).

## 2.5 Efficient Sequence Modeling and State Space Approaches

Building on the search for more efficient long-range modeling, recent Mamba-based approaches have emerged as a promising alternative to Transformers. By utilizing Selective State Space Models (SSMs), these architectures achieve linear scaling with respect to sequence length, effectively capturing long-range dependencies without the memory overhead of self-attention (Hamad et al., 2025). Specifically, the Multi-scale Wavelet-Mamba framework (Li et al., 2025) further enhances this paradigm by incorporating wavelet transforms to decompose complex traffic signals into multiple frequency components, allowing the Mamba blocks to capture both fine-grained fluctuations and global trends with high fidelity. However, a critical efficiency tradeoff remains; while Mamba architectures excel at processing long sequences, the optimization of selective scan operations and hardware-aware parallelization can introduce significant complexity in model configuration. This often results in a higher number of trainable parameters and increased optimization difficulty even with state space modeling, contrasting with the fixed-reservoir simplicity of our proposed DC-ESN.

## 2.6 Reservoir computing based Spatiotemporal Models

Despite their effectiveness, the above mentioned deep architectures often involve high computational costs and a large number of trainable parameters, a trend frequently characterized as "Red AI" (Schwartz et al., 2020). This reliance on massive computational power not only limits scalability in real-time applications but also raises significant barriers to research inclusivity (Schwartz et al., 2020). Hence, reservoir computing frameworks such as the Echo State Network (ESN) (Jaeger, 2001) and its deep variants (Gallicchio et al.,

2018) (Gallicchio & Micheli, 2017) have gained attention for their ability to model temporal dynamics efficiently using fixed recurrent reservoirs and fast training. However, ESNs alone cannot capture spatial dependencies inherent in graph-structured traffic data. Thus, subsequent studies have increasingly explored the integration of Echo State Networks (ESNs) and Graph Neural Networks (GNNs) for spatiotemporal learning problems. (Soussia et al., 2026) proposed Reservoir-Based Graph Convolutional Networks that integrate reservoir computing principles directly into graph convolutional architectures. Reservoir computing based frameworks like GraphRC (Gallicchio & Micheli, 2010), Improved GraphRC (Han & Zhao, 2022) and ESSTPNet (Zhang & Zhang, 2025) were developed to address the computational constraints problem in traffic forecasting research. (Han & Zhao, 2022) proposed an interpretable graph reservoir computing framework with temporal pattern attention to improve interpretability in graph-based reservoir architectures. (Zhang & Zhang, 2025) introduced ESSTPNet, which combines deep ESN reservoirs with adaptive multi-scale decoding for spatiotemporal forecasting. These studies demonstrate the growing interest in graph reservoir computing as an efficient alternative to fully trainable recurrent models.

However, the proposed DC-ESN differs from prior approaches in several key aspects.

- First, DC-ESN is specifically designed as a lightweight replacement for the recurrent component of DCRNN by combining diffusion convolution with a fixed ESN reservoir inside a Seq2Seq forecasting framework.

- Second, unlike prior methods that primarily focus on graph representation learning or attention-enhanced reservoirs, DC-ESN explicitly decouples spatial learning and temporal modeling through trainable diffusion convolution and fixed sparse reservoirs, respectively.

- Third, the proposed Projection-Concatenation-Projection (PCP) diffusion mechanism preserves multi-scale forward and backward diffusion information instead of aggregating them through simple summation.

- Finally, compared with prior ESN-GNN approaches, our work emphasizes computational efficiency and Green AI considerations by substantially reducing trainable parameters, GPU memory usage, and training time while maintaining competitive forecasting accuracy (Schwartz et al., 2020).

Our work focuses on the benefits of static graph modeling, which offers superior robustness in recurring traffic scenarios. Unlike MegaCRN or PM-MemNet, which rely on complex learnable or time-varying graph structures, DC-ESN operates in a static spatial domain. This design choice prioritizes model interpretability, computational efficiency, and simplicity, maintaining high performance without the sensitivity to noise often found in dynamic modeling (Jiang et al., 2023).

Existing studies largely focus on improving predictive accuracy through increasingly sophisticated architectures. However, in many practical deployments, inference latency, memory footprint, and hardware constraints are also critical. This motivates the central question of our work:

> Can competitive spatiotemporal traffic forecasting be achieved using substantially simpler recurrent dynamics than gated RNNs and modern high-complexity architectures?

To answer this question, we propose DC-ESN, which combines diffusion convolution for spatial modeling with an Echo State Network for efficient temporal dynamics.

## 3 Methodology

In this study, we propose the Diffusion Convolutional Echo State Network (DC-ESN), a hybrid deep learning framework designed for spatiotemporal traffic forecasting. Our approach synergizes the spatial feature extraction capabilities of Graph Neural Networks (GNNs), specifically Diffusion Convolution (Li et al., 2018), with the computational efficiency of Reservoir Computing (Jaeger, 2001). Unlike the standard DCRNN, which relies on fully trainable Gated Recurrent Units (GRUs), the DC-ESN employs a large, fixed sparse reservoir to model temporal dynamics, restricting gradient-based training to the spatial diffusion and readout modules.

### 3.1 Problem Formulation

Consider a directed graph $\mathcal{G} = (\mathcal{V}, \mathcal{E}, \boldsymbol{W})$ with $|\mathcal{V}| = N$ nodes, $\mathcal{E}$ denotes the edge set between nodes and $\boldsymbol{W} \in \mathbb{R}^{N \times N}$ denotes a weighted adjacency matrix describing pairwise relationships between nodes. A graph signal at time $t$ is represented as $\boldsymbol{X}^{(t)} \in \mathbb{R}^{N \times F}$, where each node is associated with $F$ features. In the context of traffic forecasting, nodes correspond to traffic sensors distributed across a road network, and the graph structure encodes spatial connectivity between sensors (e.g., road distance or topology) (Li et al., 2018). The graph signal $\boldsymbol{X}^{(t)}$ therefore represents traffic measurements such as speed, flow, or occupancy collected at time $t$.

Given a sequence of historical graph signals

$$\{\boldsymbol{X}^{(t-T'+1)}, \dots, \boldsymbol{X}^{(t)}\},$$

the objective is to learn a predictive mapping

$$\mathcal{F} : (\boldsymbol{X}^{(t-T'+1)}, \dots, \boldsymbol{X}^{(t)}, \mathcal{G}) \to (\boldsymbol{X}^{(t+1)}, \dots, \boldsymbol{X}^{(t+T)}),$$

which forecasts the future evolution of the graph signal over the next $T$ time steps.

This formulation captures the spatiotemporal structure of traffic data, where spatial dependencies are governed by the graph topology and temporal dependencies arise from the dynamic evolution of traffic conditions. For notation see Table 6 in Appendix.

### 3.2 Spatial Modeling via Graph Diffusion

To characterize the non-Euclidean spatial dependencies inherent in traffic sensor networks, we adopt a graph-based diffusion framework (Li et al., 2018). By representing the sensor topology as a directed graph $\mathcal{G}$, we can effectively model the asymmetric propagation of traffic flow. We define this spatial influence through a random walk with a restart probability $\gamma \in [0, 1]$, where the transition dynamics are governed by the out-degree normalized adjacency matrix $\boldsymbol{D}_O^{-1}\boldsymbol{W}$. Here, $\boldsymbol{D}_O = \text{diag}(\boldsymbol{W}\boldsymbol{1})$ serves as the out-degree diagonal matrix, with $\boldsymbol{1} \in \mathbb{R}^N$ representing an all-one vector.

As this Markovian process reaches a steady state, it converges to a stationary distribution $\boldsymbol{P} \in \mathbb{R}^{N \times N}$. Each row $\boldsymbol{P}_{i,:}$ in this distribution quantifies the "reachability" or spatial proximity originating from node $v_i \in \mathcal{V}$. Following the theoretical foundations of random walks on graphs (Teng, 2016), this process can be expressed in the following closed form:

$$\boldsymbol{P} = \sum_{k=0}^{\infty} \gamma(1-\gamma)^k (\boldsymbol{D}_O^{-1}\boldsymbol{W})^k \tag{1}$$

In practice, the infinite series is approximated by truncating the diffusion process to $K$ steps. To ensure the model remains sensitive to both upstream and downstream dependencies, we incorporate a reciprocal diffusion component defined by the in-degree transition matrix $\boldsymbol{D}_I^{-1}\boldsymbol{W}^\top$, where $\boldsymbol{D}_I = \text{diag}(\boldsymbol{W}^\top\boldsymbol{1})$ denotes the in-degree diagonal matrix. By integrating both forward and backward diffusion processes, the architecture effectively captures the bidirectional nature of traffic information flow within the network (Li et al., 2018).

### 3.3 Diffusion Convolution Implementation

To model spatial relationships, we employ a diffusion convolution operation over graph signals (Kipf & Welling, 2016). While the theoretical formulation involves summing diffusion steps, our implementation utilizes a Projection-Concatenation-Projection (PCP) architecture to enhance spatial expressiveness and computational efficiency on GPUs.

Let $\boldsymbol{Z}^{(t)} = [\boldsymbol{X}^{(t)} \| \boldsymbol{H}^{(t-1)}] \in \mathbb{R}^{N \times (F+d_h)}$ be the concatenation of the current input $\boldsymbol{X}^{(t)} \in \mathbb{R}^{N \times F}$ and the previous recurrent state $\boldsymbol{H}^{(t-1)} \in \mathbb{R}^{N \times d_h}$ at time step $t$. The spatial processing proceeds in three stages:

1. Input Projection (Dimensionality Alignment): First, we project the high-dimensional input into the hidden dimension $d_h$ using a trainable weight matrix $\boldsymbol{W}_{\text{pre}} \in \mathbb{R}^{(F+d_h) \times d_h}$. This bottleneck step significantly reduces the parameter count for subsequent operations:

$$\tilde{\boldsymbol{Z}}^{(t)} = \boldsymbol{Z}^{(t)} \boldsymbol{W}_{\text{pre}} \tag{2}$$

2. Multi-Hop Diffusion and Concatenation: Instead of summing the diffusion steps, we explicitly compute the diffusion states for $K$ hops in both forward and backward directions and concatenate them into a comprehensive spatial vector $\boldsymbol{U}^{(t)}$ as shown in Figure 2. This allows the model to learn distinct weights for different diffusion distances (Abu-El-Haija et al., 2019) (e.g., immediate vs. distant neighbors):

$$\boldsymbol{U}^{(t)} = \left[ \tilde{\boldsymbol{Z}}^{(t)} \, \Big\|_{k=1}^{K} \, (\boldsymbol{D}_O^{-1} \boldsymbol{W})^k \tilde{\boldsymbol{Z}}^{(t)} \, \Big\|_{k=1}^{K} \, (\boldsymbol{D}_I^{-1} \boldsymbol{W}^\top)^k \tilde{\boldsymbol{Z}}^{(t)} \right] \tag{3}$$

where $\|$ denotes concatenation along the feature dimension. The resulting vector $\boldsymbol{U}^{(t)} \in \mathbb{R}^{N \times (d_h \cdot (2K+1))}$ preserves multi-scale spatial information.

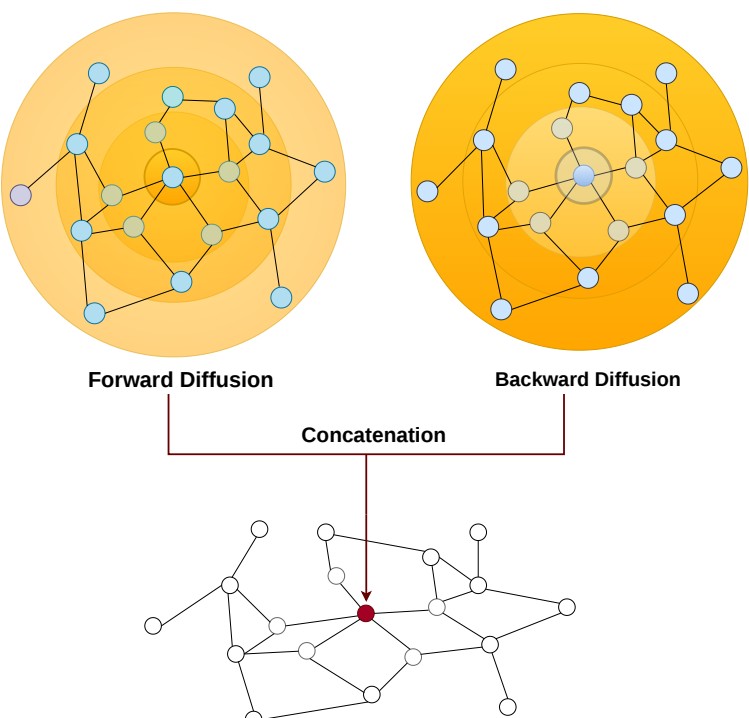

Figure 2: Diffusion convolution using MixHop to aggregate multi-hop forward and reverse random walks.

3. Spatial Mixing: Finally, the spatial output $\boldsymbol{S}^{(t)}$ is obtained by mapping the concatenated features back to the hidden dimension using a trainable post-weight matrix $\boldsymbol{W}_{\text{post}} \in \mathbb{R}^{(d_h \cdot (2K+1)) \times d_h}$:

$$\boldsymbol{S}^{(t)} = \boldsymbol{U}^{(t)} \boldsymbol{W}_{\text{post}} \tag{4}$$

## 3.4 Temporal Dependency Modeling: The DC-ESN Cell

To capture the temporal dynamics of traffic flow, we replace the fully trainable Gated Recurrent Unit (GRU) used in prior models (e.g., DCRNN) with an Echo State Network (ESN) reservoir architecture as

shown in Figure 3. Unlike traditional RNNs, ESNs maintain a large, fixed recurrent reservoir with randomly initialized connections. Only the spatial projection weights ($\boldsymbol{W}_{\text{pre}}, \boldsymbol{W}_{\text{post}}$) and the readout layer are trained,

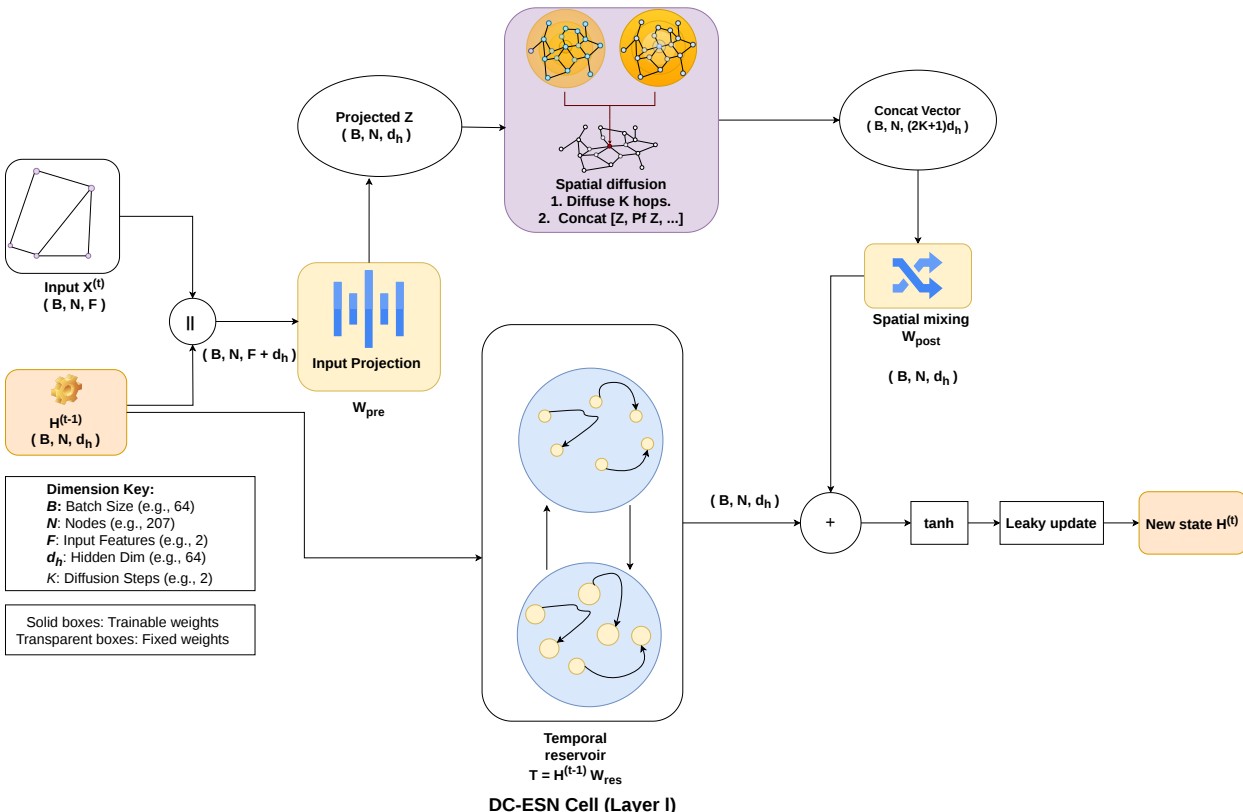

Figure 3: The DC-ESN Cell Layer l decoupling spatial and temporal modeling. The cell integrates a trainable diffusion convolution module (top) to capture spatial dependencies via multi-hop concatenation with a fixed, sparse reservoir (bottom) for efficient temporal memory.

enabling highly efficient learning while preserving strong nonlinear temporal representation capabilities.

### 3.4.1 Reservoir Dynamics

Let $\boldsymbol{X}^{(t)} \in \mathbb{R}^{N \times F}$ denote the input at time $t$. The update mechanism for the hidden state $\boldsymbol{H}^{(t-1)} \in \mathbb{R}^{N \times d_h}$ combines the trainable spatial features with the fixed temporal dynamics. The state update equation is given by:

$$\boldsymbol{H}^{(t)} = (1 - \alpha)\boldsymbol{H}^{(t-1)} + \alpha \tanh\left(\boldsymbol{S}^{(t)} + \boldsymbol{T}^{(t)}\right) \tag{5}$$

where:

- $\alpha \in (0, 1]$ is the leaking rate, acting as a low-pass filter for memory retention.
- $\boldsymbol{S}^{(t)} \in \mathbb{R}^{N \times d_h}$ is the Spatial Component. This term captures the multi-scale spatial dependencies of the current input $\boldsymbol{X}^{(t)}$ and previous state $\boldsymbol{H}^{(t-1)}$ via the PCP architecture described in subsection 3.3.

- $\boldsymbol{T}^{(t)} \in \mathbb{R}^{N \times d_h}$ is the Temporal Component. To allow for heterogeneous temporal dynamics across the network, we utilize a node-specific reservoir operation:

$$\boldsymbol{T}_i^{(t)} = \boldsymbol{H}_i^{(t-1)} \mathbf{W}_{\mathrm{res},i} \quad \forall i \in \{1, \ldots, N\} \tag{6}$$

Here, $\mathbf{W}_{\mathrm{res}} \in \mathbb{R}^{N \times d_h \times d_h}$ is a tensor containing independent reservoir weights for each node. These weights are initialized sparsely with spectral radius $\rho < 1$ and are frozen (non-trainable), ensuring the "Echo State Property" and avoiding the vanishing gradient problem.

### 3.5 Encoder–Decoder Architecture

We adopt a Sequence-to-Sequence (Seq2Seq) framework to handle multi-step forecasting. Both the encoder and decoder are constructed using stacked DC-ESN cells as depicted in Figure 4.

#### 3.5.1 Encoder

The encoder processes the historical sequence $\boldsymbol{X} = [\boldsymbol{X}^{(1)}, \ldots, \boldsymbol{X}^{(T')}]$. At each step $t$, the hidden state evolves according to the DC-ESN dynamics described above. The encoder summarizes the temporal evolution of the input sequence into the final hidden states $\boldsymbol{H}^{(T')}$, which compactly represent the past spatiotemporal context.

#### 3.5.2 Decoder

The decoder generates forecasts $\hat{\boldsymbol{X}} = [\hat{\boldsymbol{X}}^{(T'+1)}, \ldots, \hat{\boldsymbol{X}}^{(T'+T)}]$ autoregressively. Initialized with the encoder's final state, the decoder predicts the next time step. The output is projected to the target dimension via a trainable linear readout:

$$\hat{\boldsymbol{X}}^{(t)} = \boldsymbol{H}^{(t)} \boldsymbol{W}_{\mathrm{out}} + \boldsymbol{b} \tag{7}$$

where, $\boldsymbol{W}_{\mathrm{out}}$ and $\boldsymbol{b}$ are learnable output weights and bias.

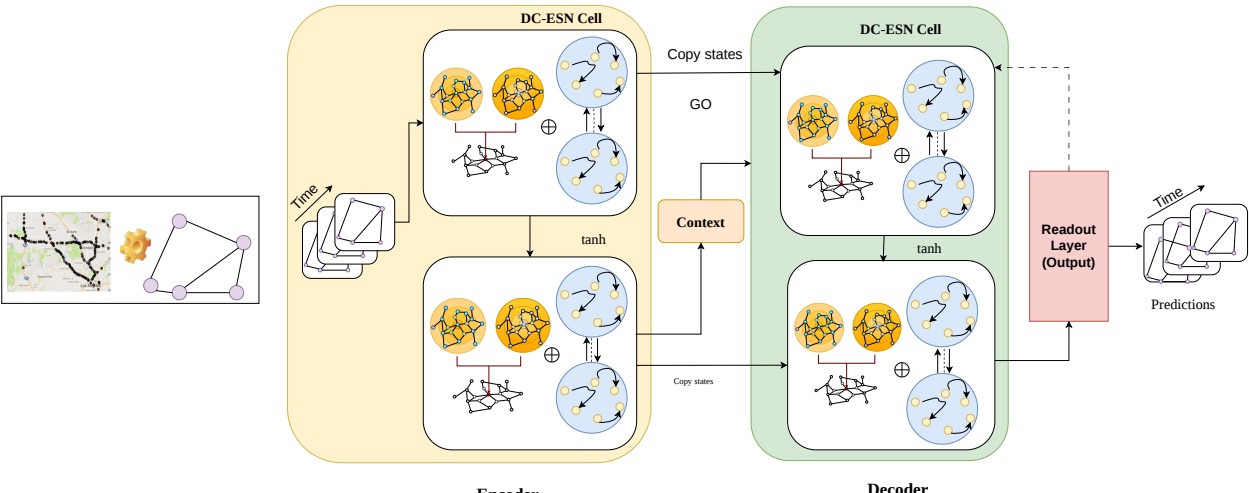

Figure 4: Model Architecture of DC-ESN

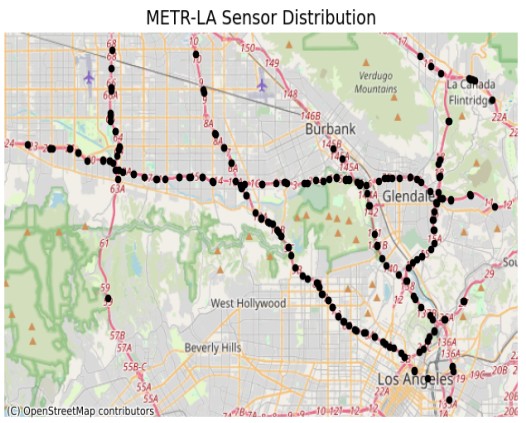 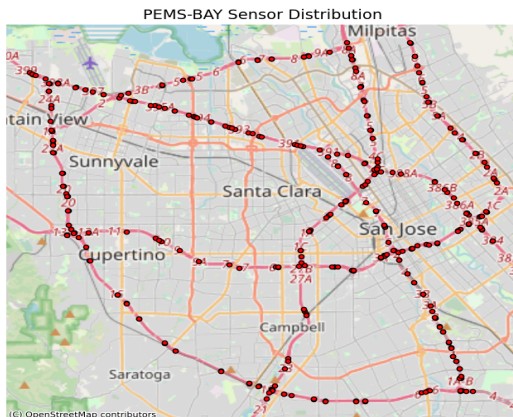

Figure 5: Sensor network of METR-LA(left) and PEMS-BAY(right) datasets.

### 3.6 Hybrid Optimization Strategy

The training process involves a hybrid optimization strategy where parameters are partitioned into fixed and trainable sets:

- Fixed: $\Theta_{\text{fix}} = \{\mathbf{W}_{\text{res}}\}$ (Reservoir weights).

- Trainable: $\Theta_{\text{train}} = \{\boldsymbol{W}_{\text{pre}}, \boldsymbol{W}_{\text{post}}, \boldsymbol{W}_{\text{out}}\}$ (Spatial and Readout weights).

We minimize the forecasting loss (e.g., Mean Absolute Error(MAE)) using backpropagation through time with respect to the trainable parameters. Gradients flow through the reservoir states to update the spatial parameters $\Theta_{\text{train}}$, but do not update $\mathbf{W}_{\text{res}}$. To mitigate exposure bias, the discrepancy between training (using ground truth) and inference (using model predictions), we employ scheduled sampling (teacher forcing). At iteration $i$, the decoder receives the ground truth with probability $\epsilon_i$ or its own prediction with probability $(1 - \epsilon_i)$. The probability $\epsilon_i$ decays from 1 to 0 during training, gradually adapting the model to the inference distribution.

## 4 Datasets

### 4.1 Datasets and Preprocessing

The DC-ESN framework was evaluated on two large-scale, real-world traffic benchmarks: METR-LA and PEMS-BAY (Li et al., 2018). The METR-LA data set consists of traffic speed data captured by 207 loop detector sensors on Los Angeles County highways, spanning a four-month period from March 1, 2012, to June 30, 2012 (Jagadish et al., 2014). The PEMS-BAY dataset comprises traffic speed data from 325 sensors in the Bay Area, collected by the California Department of Transportation (CalTrans) over six months, from January 1, 2017, to May 31, 2017. For both datasets, traffic readings were aggregated into 5-minute windows and normalized using Z-Score normalization, with a chronological data split of 70% for training, 10% for validation and 20% for testing. Table 1 summarizes the datasets used in the experiments and Figure 5 shows the sensor distribution of both the datasets.

Table 1: Summary of datasets used in experiments

| Dataset | Sensors | Time Steps | Time Range | Sample Rate | Features |
|---------|---------|------------|------------|-------------|----------|
| METR-LA | 207 | $\approx 34{,}272$ | Mar 1, 2012 – Jun 30, 2012 | 5 min | 1 (Speed) |
| PEMS-BAY | 325 | $\approx 52{,}116$ | Jan 1, 2017 – May 31, 2017 | 5 min | 1 (Speed) |

### 4.2 Graph Construction

To incorporate the spatial topology of the road network, the sensor network was modeled as a weighted directed graph. The adjacency matrix weights, $W_{ij}$, were calculated using a thresholded Gaussian kernel based on the road network distance between sensors (Shuman et al., 2013). The weight calculation is defined as:

$$W_{ij} = \begin{cases} \exp\left(-\frac{\text{dist}(v_i, v_j)^2}{\sigma^2}\right) & \text{if } \text{dist}(v_i, v_j) \leq \kappa, \\ 0 & \text{otherwise,} \end{cases} \tag{8}$$

where $W_{ij}$ represents the edge weight between sensors $v_i$ and $v_j \in \mathcal{V}$, $\text{dist}(v_i, v_j)$ is the road network distance, $\sigma$ is the standard deviation of distances, and $\kappa$ is the threshold. To compute $\text{dist}(v_i, v_j)$ for each node pair $v_i$, $v_j$ of the sensor network for METR-LA and PEMS-BAY datasets, each traffic sensor is provided with fixed geographic coordinates (latitude and longitude). These coordinates were first standardized and converted into a geospatial representation. To obtain realistic spatial relationships, the drivable road networks for Los Angeles (METR-LA) and the San Francisco Bay Area (PEMS-BAY) were extracted from OpenStreetMap using the OSMnx library. The resulting transportation networks were represented as directed, weighted graphs in which edges correspond to road segments and weights denote their physical lengths. Each sensor was mapped to its nearest node in the corresponding road graph using a spatial nearest-neighbor search. Road-network distances between sensors were then computed using Dijkstra's shortest-path algorithm, ensuring that inter-sensor distances reflect true drivable routes rather than Euclidean separation. These distances were subsequently used to construct the adjacency matrix for graph-based traffic forecasting models. This preprocessing step ensures that the spatial structure accurately captures the real transportation network in both regions.

## 5 Experimental Results and Discussions

### 5.1 Baseline Comparisons

The performance of DC-ESN was benchmarked against a diverse set of existing models, including traditional statistical methods such as Historical Average (HA), ARIMA with Kalman filtering, Vector Auto-Regression (VAR) (Hamilton, 1994). Additionally, deep learning baselines were employed, specifically Feed Forward Neural Networks (FNN) and Recurrent Neural Networks utilizing fully connected LSTM units (FC-LSTM) and spatiotemporal model mainly DCRNN that uses Diffusion convolution for spatial and GRU for temporal dependency modeling (Sutskever et al., 2014). Evaluation metrics included Mean Absolute Error (MAE), Mean Absolute Percentage Error (MAPE), and Root Mean Squared Error (RMSE) across 15-minute, 30-minute, and 1-hour forecasting horizons.

### 5.2 Baseline Implementation Details

To validate the performance of our approach, we compare it against a comprehensive set of baselines ranging from statistical methods to deep learning frameworks. The hyperparameters were selected using the "Tree-structured Parzen Estimator" (Bergstra et al., 2011). The details of configurations for each model are as follows:

- Historical Average (HA): This method models traffic as a seasonal process. It predicts future speeds by calculating the weighted average of the same time slot from previous weeks. Specifically, we use a period of one week and average the data from the preceding four weeks (e.g., predicting a Wednesday 5:00 PM slot using the average of the last four Wednesdays at 5:00 PM).

- ARIMA$_{kal}$: Auto-Regressive Integrated Moving Average model with Kalman filtering, with model order $(3, 0, 1)$.

- Vector Auto-Regression (VAR): This architecture captures pairwise relationships between sensors. We set the number of time lags to 3, following standard time series analysis approaches (Hamilton, 1994).

- Feed Forward Neural Network (FNN): We employ a fully connected neural network consisting of two hidden layers with 256 units each. Following the experimental configuration in (Li et al., 2018), dropout with probability 0.5 and L2 regularization with weight decay $1 \times 10^{-2}$ are applied to the hidden layers. The model is trained using the Adam optimizer (Abadi et al., 2016) with a batch size of 64 and MAE as the loss function. The learning rate is initialized at $1 \times 10^{-3}$ and decays by a factor of 0.1 every 20 epochs starting from epoch 50.

- FC-LSTM: We adopt a sequence-to-sequence architecture based on Long Short-Term Memory (LSTM) units with peephole connections (Sutskever et al., 2014). Following the experimental configuration in (Li et al., 2018), both the encoder and decoder contain two recurrent layers with 256 hidden units each. L1 and L2 regularization with coefficients $2 \times 10^{-5}$ and $5 \times 10^{-4}$ are applied to the model parameters. Training is performed using a batch size of 64 with MAE as the loss function. The learning rate is initialized at $1 \times 10^{-4}$ and decayed by a factor of 0.1 every 10 epochs starting from epoch 20.

- Diffusion Convolutional Recurrent Neural Network (DCRNN): We also include DCRNN (Li et al., 2018) as a strong baseline. The architecture uses an encoder-decoder structure with two recurrent layers, each containing 64 units. The maximum steps for the random walk in the diffusion convolution ($K$) is set to 3. Training utilizes scheduled sampling (Bengio et al., 2015) with a reverse sigmoid decay $\epsilon_i = \tau/(\tau + \exp(i/\tau))$, where $\tau = 3000$. The initial learning rate is $1 \times 10^{-2}$, decaying by a factor of 0.1 every 10 epochs starting from the 20th epoch.

- PM-MemNet: A memory-augmented neural network that captures long-term traffic prototypes using a persistent memory bank. The architecture utilizes a $k$-nearest neighbor ($k = 3$) lookup with a cosine similarity function to retrieve contextually relevant historical patterns. Following the original design, the model employs a hidden dimension of 128, three diffusion convolution layers ($L = 3$) with 3 hops each, and Xavier initialization. For consistency with our other benchmarks, we adopt the $T = 12$ sequence setting. Optimization is performed using the Adam optimizer with a batch size of 64 and an initial learning rate of 0.001, subject to multi-step decay at epochs 50 and 60. The integration of a prior knowledge matrix ensures the addressing mechanism effectively retrieves globally representative traffic prototypes for both METR-LA and PEMS-BAY datasets. (Lee et al., 2021)

- STAEformer: A state-of-the-art Spatio-Temporal Adaptive Embedding Transformer. For METR-LA and PEMS-BAY, the model consists of 3 layers with 4 attention heads each. It utilizes an 80-dimensional adaptive embedding and 24-dimensional temporal embeddings (Time-of-Day and Day-of-Week). The architecture employs a feed-forward dimension of 256 and is optimized using a batch size of 16 and a learning rate of 0.001. (Liu et al., 2023)

For all deep learning models (FNN, FC-LSTM, and DCRNN), we employ early stopping based on the validation error to prevent overfitting.

### 5.3 DC-ESN performance: Results and Ablation Studies

We evaluated the proposed DC-ESN on two benchmark traffic datasets: METR-LA (207 sensors) and PEMS-BAY (325 sensors). We compared the performance against the state-of-the-art DCRNN baseline in terms of prediction accuracy, convergence speed, and computational resource consumption. Experimental results demonstrate that DC-ESN consistently outperforms standard baselines like HA, VAR, and FC-LSTM. While high-capacity models such as STAEformer (Liu et al., 2023) and PM-MemNet (Lee et al., 2021) achieve marginally lower error rates, DC-ESN maintains competitive parity with DCRNN while utilizing significantly fewer trainable parameters and computational resources. This performance advantage is particularly robust in long-term (1-hour) forecasting." (Li et al., 2018). Table 2 and Table 3 shows the comparison of DC-ESN

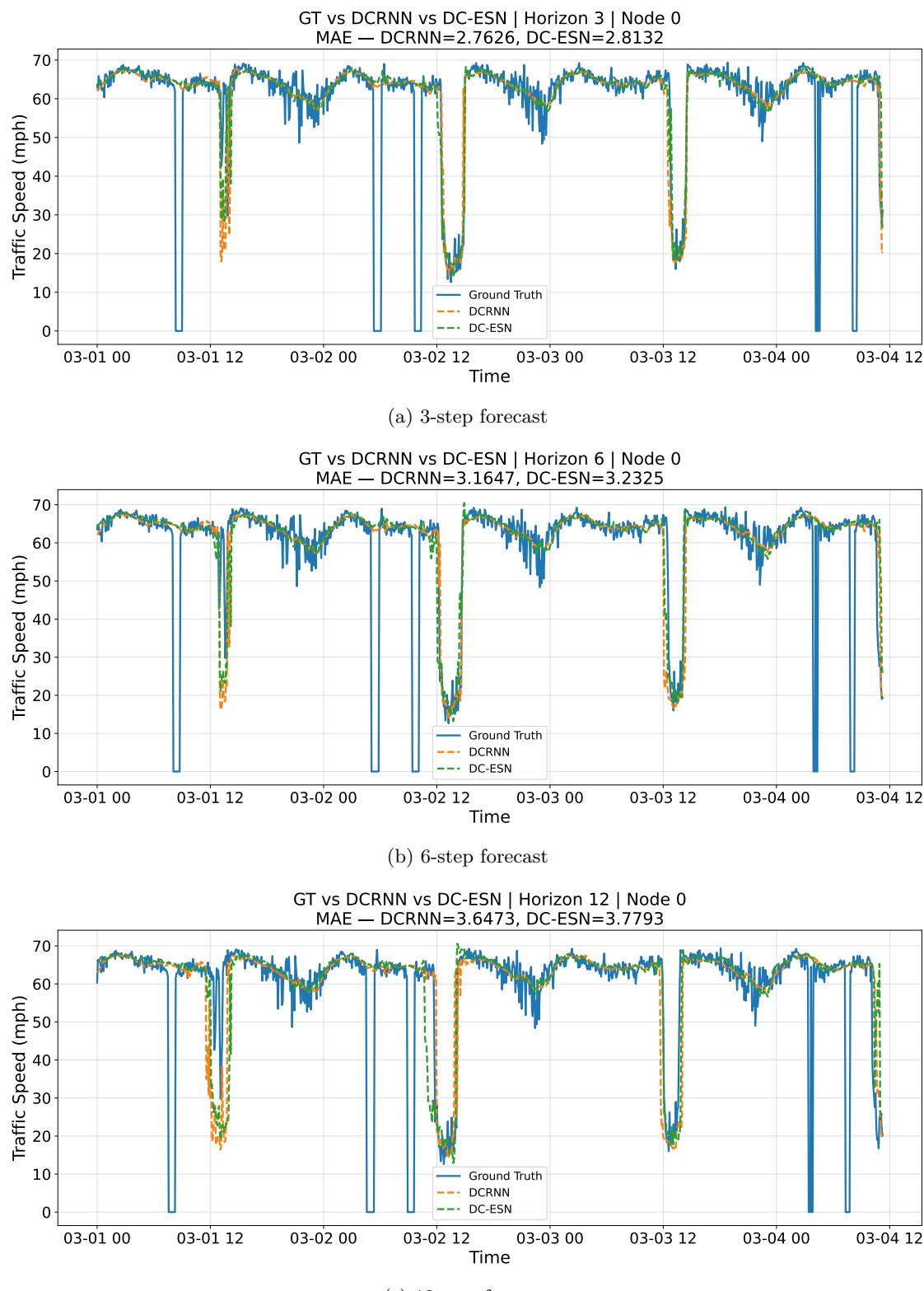

(a) 3-step forecast

(b) 6-step forecast

(c) 12-step forecast

Figure 6: Forecast visualization of DC-ESN and DCRNN on the METR-LA dataset across 3-, 6-, and 12-step prediction horizons.

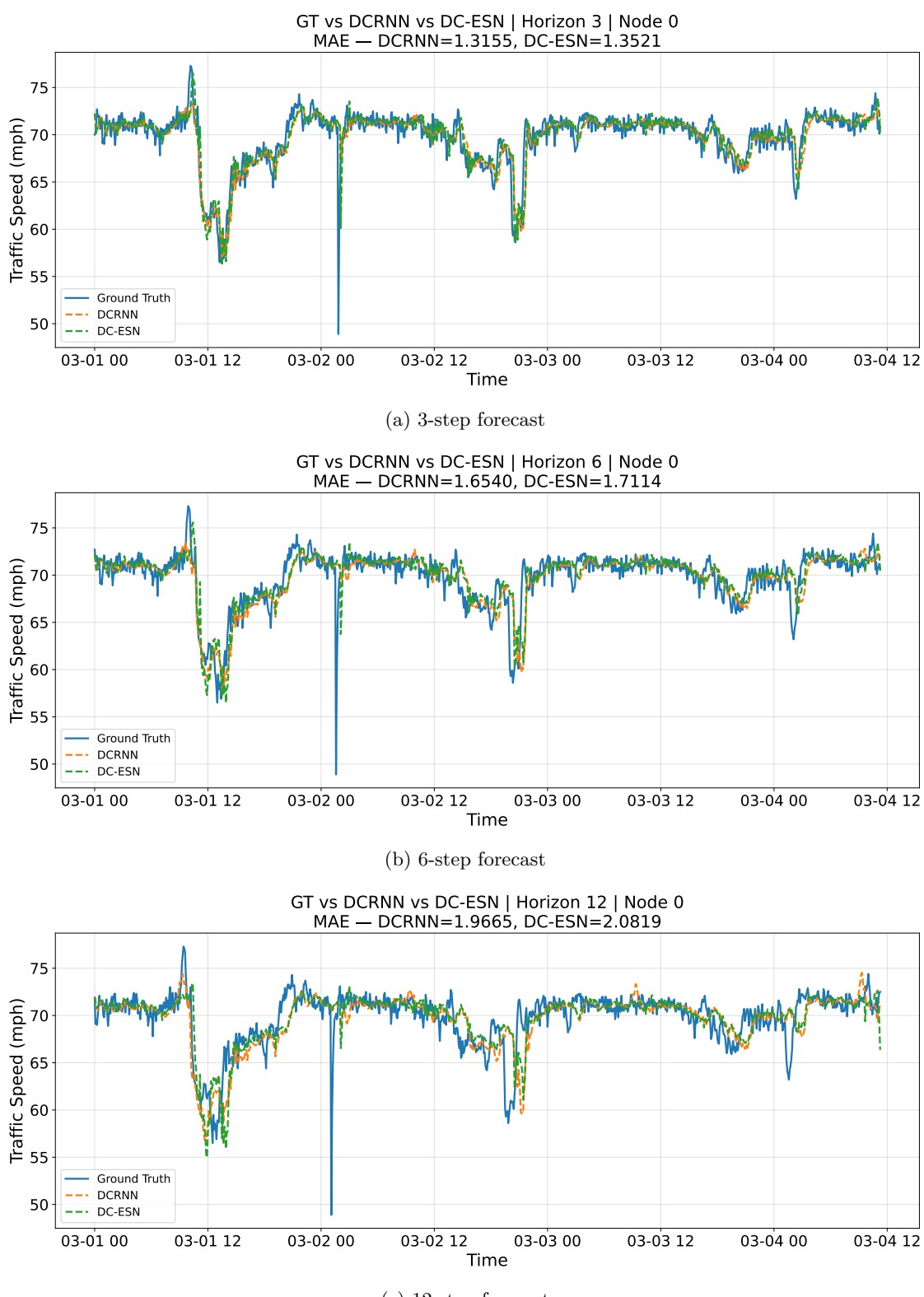

(a) 3-step forecast

(b) 6-step forecast

(c) 12-step forecast

Figure 7: Forecast visualization of DC-ESN and DCRNN on the PEMS-BAY dataset across 3-, 6-, and 12-step prediction horizons. Ground truth and predicted traffic speed trajectories are shown.

with baseline for different seeds (42, 100, 199, 500) as well as for single seed reservoir initialized randomly for datasets METR-LA and PEMS-BAY respectively . Figure 8 and Figure 9 show the performance comparison of DC-ESN (for both the datasets) against DCRNN in terms of mean $\pm$ standard deviation of metrics for MAE, RMSE, and MAPE across all forecasting horizons for different seeds 42, 100, 199, 500. Figure 6 and Figure 7 are forecast visualization of DC-ESN and DCRNN on both the dataset (METR-LA, PEMS-BAY) across 3-, 6-, and 12-step prediction horizons.

Table 2: Performance comparison on the METR-LA benchmark. Results are reported as mean $\pm$ std. Best values are shown in bold.

| Model | METR-LA | | | | | | | | |
| | 15 min | | | 30 min | | | 1 hour | | |
| | MAE | RMSE | MAPE | MAE | RMSE | MAPE | MAE | RMSE | MAPE |
| HA | 4.15 | 7.77 | 12.90% | 4.15 | 7.77 | 12.90% | 4.15 | 7.77 | 12.90% |
| ARIMA$_{kal}$ | 4.02 | 8.69 | 9.39% | 5.09 | 11.13 | 12.75% | 6.79 | 14.21 | 16.71% |
| VAR | 4.37 | 7.78 | 10.10% | 5.40 | 9.37 | 12.70% | 6.50 | 10.68 | 15.84% |
| FNN (Li et al., 2018) | 3.99 | 7.94 | 9.9% | 4.23 | 8.17 | 12.9% | 4.49 | 8.69 | 14.0% |
| FC-LSTM (Li et al., 2018) | 3.44 | 6.30 | 9.6% | 3.77 | 7.23 | 10.9% | 4.37 | 8.69 | 13.2% |
| **DC-ESN (Mean ± std)** | 2.830 ± 0.03 | 5.395 ± 0.08 | 7.36% ± 0.04 | 3.257 ± 0.05 | 6.493 ± 0.10 | 8.94% ± 0.10 | 3.821 ± 0.07 | 7.803 ± 0.13 | 11.05% ± 0.21 |
| DC-ESN (Single Seed) | 2.81 | 5.35 | 7.31% | 3.23 | 6.44 | 8.84% | 3.78 | 7.74 | 10.86% |
| DCRNN | 2.763 | 5.313 | 7.14% | 3.165 | 6.388 | 8.42% | 3.647 | 7.574 | 10.00% |
| PM-MemNet | 2.892 | 5.665 | 7.75% | 3.282 | 6.793 | 9.30% | 3.699 | 7.860 | 10.96% |
| STAEformer | **2.634** | **5.086** | **6.84%** | **2.963** | **6.027** | **8.18%** | **3.337** | **7.033** | **9.80%** |

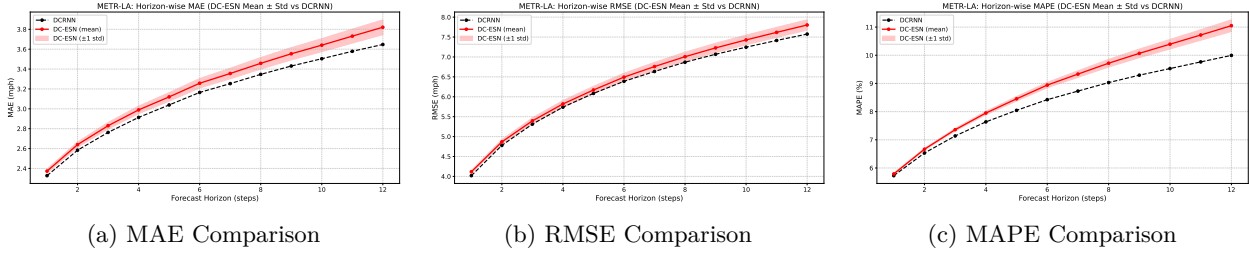

(a) MAE Comparison        (b) RMSE Comparison        (c) MAPE Comparison

Figure 8: Performance comparison of DC-ESN (for METR-LA dataset) against DCRNN in terms of mean $\pm$ standard deviation of metrics for MAE, RMSE, and MAPE across all forecasting horizons for different seeds 42, 100, 199, 500.

### 5.4 Performance Analysis

Empirical testing was conducted on an NVIDIA RTX 6000 Ada Generation GPU.

**1. Computational Efficiency and Speedup:** The proposed DC-ESN demonstrates a significant acceleration in training performance compared to the gated DCRNN baseline. On the METR-LA dataset, the average training latency per epoch was reduced from ∼87.4s to ∼17.8s, representing a ∼4.9× speedup. This efficiency gain is further amplified on the larger PEMS-BAY dataset, where the epoch time dropped from ∼164.1s to ∼ 27.8s this shows almost 5.9× improvement. This time complexity reduction comes from the structural design of the Echo State Network; while the model utilizes Backpropagation Through Time

Table 3: Performance comparison on the PEMS-BAY benchmark. Results are reported as mean ± std. Best values are shown in bold.

| Model | PEMS-BAY | | | | | | | | |
| | 15 min (H3) | | | 30 min (H6) | | | 1 hour (H12) | | |
| | MAE | RMSE | MAPE | MAE | RMSE | MAPE | MAE | RMSE | MAPE |
|---|---|---|---|---|---|---|---|---|---|
| HA | 2.88 | 5.59 | 6.76% | 2.88 | 5.59 | 6.76% | 2.88 | 5.59 | 6.76% |
| ARIMA$_{kal}$ | 1.60 | 3.43 | 3.59% | 2.18 | 4.98 | 4.65% | 3.05 | 7.02 | 6.84% |
| VAR | 1.74 | 3.09 | 3.60% | 2.33 | 4.15 | 5.02% | 2.92 | 5.11 | 6.46% |
| FNN (Li et al., 2018) | 2.20 | 4.42 | 5.2% | 2.30 | 4.63 | 5.4% | 2.46 | 4.98 | 5.9% |
| FC-LSTM (Li et al., 2018) | 2.05 | 4.19 | 4.8% | 2.20 | 4.55 | 5.2% | 2.37 | 4.96 | 5.7% |
| **DC-ESN (mean ± std)** | 1.354 ± 0.00 | 2.830 ± 0.01 | 2.85% ± 0.01 | 1.714 ± 0.01 | 3.874 ± 0.02 | 3.92% ± 0.04 | 2.091 ± 0.01 | 4.801 ± 0.03 | 5.04% ± 0.05 |
| **DC-ESN (Single Seed)** | 1.35 | 2.82 | 2.85% | 1.71 | 3.87 | 3.93% | 2.08 | 4.79 | 5.06% |
| DCRNN | 1.316 | 2.770 | 2.75% | 1.654 | 3.778 | 3.75% | 1.966 | 4.611 | 4.71% |
| PM-MemNet | 1.359 | 2.848 | 2.89% | 1.680 | 3.784 | 3.78% | 1.992 | 4.563 | 4.62% |
| STAEformer | **1.297** | **2.768** | **2.73%** | **1.594** | **3.656** | **3.58%** | **1.848** | **4.285** | **4.37%** |

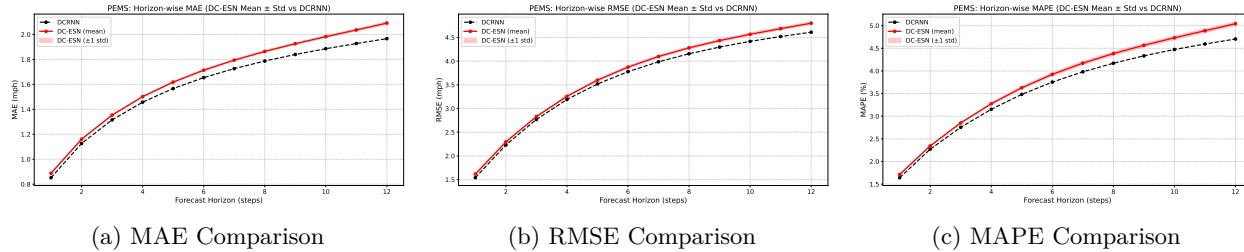

(a) MAE Comparison  (b) RMSE Comparison  (c) MAPE Comparison

Figure 9: Performance comparison of DC-ESN (for PEMS-BAY dataset) against DCRNN in terms of mean ± standard deviation of metrics for MAE, RMSE, and MAPE across all forecasting horizons for different seeds 42, 100, 199, 500.

(BPTT) for optimization, the recurrent reservoir weights ($\mathbf{W}_{res}$) remain frozen. By restricting gradient computation to the spatial diffusion and output projection layers, this architecture bypasses the complex derivative chains required to train internal gates, significantly reducing the backpropagation overhead. Also notably, DC-ESN trains ∼4× faster than STAEformer and ∼5.4× faster than PM-MemNet on the same hardware, underscoring its efficiency in high-throughput environments.

**2. Memory Footprint and Edge Viability:** The DC-ESN architecture exhibits a substantially lower VRAM requirement, aligning with the "Green AI" objective of reducing the computational price tag of deep learning research (Schwartz et al., 2020). On the PEMS-BAY dataset ($N = 325$), the peak GPU memory usage dropped from 5.55 GB to 1.99 GB, representing a 64% reduction (see Table 4). In comparison, PM-MemNet requires 10.35 GB of VRAM for the same dataset, representing an 80% reduction in memory overhead for our proposed architecture. Such a reduced footprint makes it feasible to deploy models on resource-constrained edge devices where standard Transformer-based or memory-augmented models would fail due to memory overflow. Such a reduced footprint makes it feasible to deploy and retrain the model on resource-constrained edge devices or consumer-grade GPUs, where the standard DCRNN would otherwise necessitate significantly smaller batch sizes or fail due to memory overflow.

Table 4: Comparison of Computational Efficiency and Structural Scalability

| Dataset | Metric | DCRNN | PM-MemNet | STAEformer | DC-ESN (Ours) | Improvement[a] |
|---------|--------|-------|-----------|------------|---------------|----------------|
| **METR-LA** | Trainable Params | 372,353 | 1.04M | 1.26M | **106,753** | **71.3% Reduc.** |
| | Peak GPU VRAM | 3,545 MB | 6,468 MB | 2,180 MB | **1,273 MB** | **64.1% Lower** |
| | Time per Epoch | ∼88 s | ∼49 s | ∼71 s | **∼18 s** | **∼4.9× Faster** |
| **PEMS-BAY** | Trainable Params | 372,353 | 1.13M | 1.37M | **106,753** | **71.3% Reduc.** |
| | Peak GPU VRAM | 5,554 MB | 10,349 MB | 4,010 MB | **1,997 MB** | **64.0% Lower** |
| | Time per Epoch | ∼164 s | ∼150 s | ∼110 s | **∼27 s** | **∼5.9× Faster** |

*[a] All improvement metrics are calculated relative to the DCRNN baseline.*

**3. Predictive Performance and Stability:** A definitive advantage of the Reservoir Computing paradigm is the stability of temporal dynamics. DC-ESN achieves a stable plateau in all experiments. As summarized in Table 5, the DC-ESN retains over 97% of the predictive accuracy of the gated baseline while ensuring consistent performance between training and testing phases with minimal overfitting, indicating stable generalization behavior across datasets.

Table 5: Predictive Performance Comparison on METR-LA and PEMS-BAY

| Dataset | Model | Nodes | Best Val. MAE | Final Val. MAE | Stability |
|---------|-------|-------|---------------|----------------|-----------|
| METR-LA | DCRNN | 207 | **2.822** | 4.105 | |
| | **DC-ESN** | 207 | 2.897 | **3.106** | **Stable** |
| PEMS-BAY | DCRNN | 325 | **1.628** | 2.425 | |
| | **DC-ESN** | 325 | 1.706 | **1.707** | **Stable** |

**4. Accuracy vs. Efficiency Trade-off:** The results suggest that the fixed reservoir serves as a robust "pre-wired" temporal feature extractor. While high-capacity architectures like STAEformer and PM-MemNet achieve marginally lower absolute error rates, DC-ESN maintains competitive parity with the standard DCRNN baseline. The slight MAE trade-off (approx. 2.3%) is accompanied by an order of magnitude increase in training speed. Specifically, the DC-ESN achieves a 6.1× speedup over the gated DCRNN and reduces VRAM overhead by over 80% compared to PM-MemNet, proving the DC-ESN to be a suitable candidate for real-time applications and sustainable AI deployment in urban mobility networks.

# 6 Conclusion and Future Scope

The experiments conducted in this study confirm that the proposed Diffusion Convolutional Echo State Network (DC-ESN) is a lightweight, high-speed alternative to gated recurrent architectures and high-capacity transformers. Although models such as STAEformer and PM-MemNet achieve marginally higher absolute accuracy, DC-ESN successfully captures complex spatiotemporal dependencies with a significantly reduced computational overhead, making it a strong candidate for large-scale traffic network monitoring tasks where computational resources are limited.

The proposed architecture provides the distinct advantage of combining rigorous graph-based spatial modeling with lightweight, reservoir-based temporal dynamics. By decoupling the spatial learning from the temporal recurrence, we achieve significant gains in training efficiency without sacrificing the ability to model network topology.

**Future Directions**

Future work can expand on this framework in three key areas:

- **Scalability :** Testing the scalability of the proposed framework on larger sensor networks with tens of thousands of sensors without slowing down.

- **Graph Echo State Networks (GESN):** Instead of separating spatial and temporal modules, future iterations could integrate network topology directly into the reservoir construction, allowing the reservoir structure to mirror the road network.

- **Symmetry-Aware Modeling :** Investigating ESNs that explicitly model the inherent symmetries in traffic data (e.g., periodicities and reversible flows) could enhance both predictive robustness and interpretability.

- **TinyML Deployment :** The sparse, fixed nature of the DC-ESN makes it ideal for ultra-low-power edge devices. Future research will explore deploying quantized models on microcontrollers and enabling efficient on-device adaptation by retraining only the lightweight readout layer to handle local concept drift.

## Broader Impact Statement

This work aims to improve computational efficiency in spatiotemporal forecasting, enabling sustainable AI deployment in intelligent transportation systems. Reduced GPU usage and faster training may lower energy consumption in large-scale urban monitoring systems. However, forecasting systems must be deployed carefully to avoid reinforcing biased mobility policies or inequitable infrastructure planning.

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

# A  Appendix

Table 6: Summary of notation used in this work.

| Symbol | Description |
|---|---|
| $\mathcal{G} = (\mathcal{V}, \mathcal{E}, \boldsymbol{W})$ | Directed graph with vertices $\mathcal{V}$, edges $\mathcal{E}$, and weight matrix $\boldsymbol{W}$ |
| $N$ | Number of nodes (sensors) in the traffic network |
| $F$ | Number of input features per node |
| $\boldsymbol{X}^{(t)} \in \mathbb{R}^{N \times F}$ | Input signal matrix at time step $t$ |
| $d_h$ | Dimension of the Echo State Network reservoir (hidden state) |
| $\boldsymbol{H}^{(t)} \in \mathbb{R}^{N \times d_h}$ | Hidden state matrix at time step $t$ |
| $\gamma$ | Restart probability for the diffusion process |
| $\alpha$ | Leaky rate |
| $\boldsymbol{P}$ | Stationary distribution of the diffusion process |
| $K$ | Number of diffusion steps |
| $\mathbf{W}_{\text{res}}$ | Fixed reservoir weight matrix |
| $\boldsymbol{D}_O$ (and $\boldsymbol{D}_I$) | The out-degree (and in-degree) diagonal matrices |
| $\hat{\boldsymbol{X}}^{(t)}$ | Predicted signals |
| $\boldsymbol{W}_{in}$ and $\boldsymbol{W}_{out}$ | Learnable projection weights |

## A.1  Evaluation Metrics and Loss Formulation

To ensure a rigorous and unbiased comparison, the proposed DC-ESN is evaluated using the same method-ological framework as the baseline DCRNN. We assess predictive accuracy across a 12-step temporal horizon (5 to 60 minutes) using three industry-standard metrics: Mean Absolute Error (MAE), Root Mean Squared Error (RMSE), and Mean Absolute Percentage Error (MAPE).

### A.1.1  Standard Masking for Missing Values

Traffic sensor datasets often contain missing entries or recording artifacts, typically represented as zero values. To prevent these null entries from skewing the results—particularly for the MAPE calculation, where a zero value would lead to numerical divergence—we employ the standard masking strategy used in the DCRNN framework. We define the set of observed entries $\Omega$ as:

$$\Omega = \{i \mid y_i \neq 0\} \tag{9}$$

. This ensures that only valid, physically recorded traffic states contribute to both the model's optimization and the final error reporting.

### A.1.2  Masked Objective Function

The model is optimized using a masked Mean Absolute Error (MAE) loss function, defined as:

$$\mathcal{L}(y, \hat{y}) = \frac{1}{|\Omega|} \sum_{i \in \Omega} |y_i - \hat{y}_i| \tag{10}$$

By applying the mask $\Omega$ during the backward pass of the Backpropagation Through Time (BPTT) process, we ensure that gradients are computed only from valid data points. While this loss formulation is identical to the baseline, the DC-ESN demonstrates good training stability, avoiding the catastrophic gradient failures observed in gated recurrent architectures.

### A.1.3  Mathematical Formulations

for given ground truth $y_i$ and predicted state $\hat{y}_i$, the metrics are formulated as:

1. **Mean Absolute Error (MAE):**

$$\text{MAE} = \frac{1}{|\Omega|} \sum_{i \in \Omega} |y_i - \hat{y}_i| \tag{11}$$

2. **Root Mean Squared Error (RMSE):**

$$\text{RMSE} = \sqrt{\frac{1}{|\Omega|} \sum_{i \in \Omega} (y_i - \hat{y}_i)^2} \tag{12}$$

3. **Mean Absolute Percentage Error (MAPE):**

$$\text{MAPE} = \frac{1}{|\Omega|} \sum_{i \in \Omega} \left| \frac{y_i - \hat{y}_i}{y_i} \right| \times 100\% \tag{13}$$

# B  Appendix

### B.1  Baseline Comparison Table

Table 7: Performance comparison on METR-LA and PEMS-BAY benchmark with DC-ESN(single seed). Best values are shown in bold.

| Dataset | Model | 15 min | | | 30 min | | | 1 hour | | |
|---|---|---|---|---|---|---|---|---|---|---|
| | | MAE | RMSE | MAPE | MAE | RMSE | MAPE | MAE | RMSE | MAPE |
| **METR-LA** | GWNet (Liu et al., 2023) | 2.69 | 5.15 | 6.99% | 3.08 | 6.20 | 8.47% | 3.51 | 7.28 | 9.96% |
| | MegaCRN (Liu et al., 2023) | **2.52** | **4.94** | **6.44%** | **2.93** | 6.06 | **7.96%** | **3.38** | 7.23 | **9.72%** |
| | GMAN (Liu et al., 2023) | 2.80 | 5.55 | 7.41% | 3.12 | 6.49 | 8.73% | 3.44 | 7.35 | 10.07% |
| | PD-Former (Liu et al., 2023) | 2.83 | 5.45 | 7.77% | 3.20 | 6.46 | 9.19% | 3.62 | 7.47 | 10.91% |
| | **DC-ESN(our)** | 2.81 | 5.35 | 7.31% | 3.23 | 6.44 | 8.84% | 3.78 | 7.74 | 10.86% |
| **PEMS-BAY** | GWNet (Liu et al., 2023) | 1.30 | 2.73 | 2.71% | 1.63 | 3.73 | 3.73% | 1.99 | 4.60 | 4.71% |
| | MegaCRN (Liu et al., 2023) | **1.28** | **2.72** | **2.67%** | **1.60** | **3.68** | **3.57%** | **1.88** | **4.42** | **4.41%** |
| | GMAN (Liu et al., 2023) | 1.35 | 2.90 | 2.87% | 1.65 | 3.82 | 3.74% | 1.92 | 4.49 | 4.52% |
| | PD-Former (Liu et al., 2023) | 1.32 | 2.83 | 2.78% | 1.64 | 3.79 | 3.71% | 1.91 | 4.43 | 4.51% |
| | **DC-ESN(our)** | 1.354 | 2.830 | 2.85% | 1.714 | 3.874 | 3.92% | 2.091 | 4.801 | 5.04% |

