# OpenReview forum: "DC-ESN: Diffusion Convolutional Echo State Network for Spatiotemporal Traffic Forecasting"
_TMLR — Rejected by TMLR_

### Review · Reviewer_Y3VH · 2026-04-11

**Summary Of Contributions:**

The main contribution of this work is proposing a novel efficient architecture for spatio-temporal forecasting (such as traffic forecasting) called Diffusion–Convolutional Echo State Network (DC-ESN). The diffusion convolution part is to model spatial characteristics while the Echo State Network is to efficiently model temporal relations.

Authors compare their proposed DC-ESN against 6 other methods on two datasets (METR-LA and PEMS-BAY), measuring performance with the MAE, RMSE, and MAPE as well as measuring training time and memory consumption.

DC-ESN performed better than all methods except for DCRRN. However, compared to DCRRN, their architecture used 70% less parameters, 64% less memory, and was 5x faster in training.

Strengths
- novel method
- efficient
- maintains high performance (while being more efficient in terms of speed, parameters, and memory)

Weaknesses (all relatively minor)
- lack of explanation of certain math notation
- inconsistencies in math notation
- minor typos and formatting inconsistencies

**Audience:**

Yes

**Audience Explanation:**

While the subject of this work (traffic forecasting), might not be of interest the whole TMLR audience, I believe some of TMLR readers would be interested in this work. Furthermore, I believe the proposed methodology to be potential applicable to other spatio-temporal modeling problems and thus can serve as inspiration for more efficient methods in this broader area.

**Broader Impact Concerns:**

The authors write an appropriate Broader Impact Statement.

**Claims And Evidence:**

Yes

**Claims Explanation:**

Authors compare their method against 6 baselines over two datasets using 3 evaluation metrics (MAE, RMSE, MAPE). Having various baselines and testing on separate datasets with various metrics, the claims are supported.

**Requested Changes:**

- abstract
    - "we introduces..." change to "we introduce..."
    - "a Echo State..." change to "an Echo State..."
- 2 Related Work
    - "parameters ,a" change to "parameters, a"
- 3.1 Problem Formulation
    - What is $\mathcal{E}$ in the first line mean
- 3.3 Diffusion Convolution Implementation
    - At the beginning of the second paragraph:
        - What is $H$? the same as $h$ from section 3.1
        - What is $P$, an integer for describing the shape of the matrix or the $P$ from the previous section
        - $H$ is also used for describing the shape of the matrix with $\mathbb{R}^{N \times (P+H)}$, this can be confusing
    - 1. same thing. what are P and H
    - 2. add space after "spetial vector" -> "spetial vector "
    - 3. start new paragraph for the paragraphe beginning with "3. Spatial Mixing"
- 3.4.1
    - You start with pointing out what X is, but where is X in equation 5.
    - Last paragraph also uses H for size of matrix

- 5.1 / 5.2
    - the sentence "To validate the ..." is the same at the end of 5.1 and at the beginning of 5.1
- 5.4
    - 3. the wording sentence with "near-zero" in "while ensuring a robust near-zero generalization gap", is not very clear. I would try to reword this to make it more clear on the stability benefits of the proposed method.


- Figure 1
    - Can you make this figure easier to read in black and white? Maybe use different line types as in Figure 7.

- Tables 2 and 3
    - these seem to contain almost exactly the same information (except you add s.d. for different seed for EC-ESN in table 3), is there a way to simply combine these tables? I think this would make results section more concise and clear

- Table 5
    - In the last row I see 1.706 as the "Best" and then "1.660" as the final. I'm a bit confused. Why is 1.660 not the best?

---

> ### Author Response · Authors · 2026-05-12
> **Revised Manuscript Link**
>
> We sincerely thank the reviewers for their valuable feedback. We have uploaded our revised manuscript, which addresses their comments, the following link to view the revision: https://openreview.net/forum?id=9AOUXLYlwy&noteId=9AOUXLYlwy

---

> ### Author Response · Authors · 2026-05-12
> **Summary, strengths and weaknesses**
>
> We sincerely thank Reviewer  for the careful reading of our manuscript and for the constructive feedback. We are encouraged that the reviewer recognized the novelty, efficiency, and practical significance of the proposed **Diffusion–Convolutional Echo State Network (DC-ESN)**. We especially thank for the acknowledgement that the proposed framework achieves competitive forecasting performance while substantially reducing training time, parameter count, and memory consumption compared to recurrent diffusion-based architectures such as DCRNN.
>
> ---
>
> ## Summary and Strengths
>
> **Reviewer Comment:**
> *The main contribution of this work is proposing a novel efficient architecture for spatio-temporal forecasting called Diffusion–Convolutional Echo State Network (DC-ESN). The diffusion convolution component models spatial characteristics, while the Echo State Network efficiently captures temporal relations. The proposed model achieves competitive performance while significantly reducing parameters, memory usage, and training time compared to DCRNN.*
>
> **Author response:**
> We thank the reviewer for the accurate summary of our work and for recognizing the importance of the efficiency–performance trade-off achieved by DC-ESN. Our primary objective is to develop a computationally efficient spatio-temporal forecasting framework that preserves strong predictive capability while substantially reducing training complexity and resource requirements. We are pleased that the reviewer found the experimental evidence convincing and well supported.
>
> ---
>
> ## Weaknesses (all relatively minor as suggested by the reviewer)
>
> * Lack of explanation of certain math notation
> * Inconsistencies in math notation
> * Minor typos and formatting inconsistencies
>
> We have carefully addressed all comments and suggestions raised by the reviewer. All corresponding changes have been incorporated into the revised manuscript. Our detailed responses are provided below:

---

> ### Author Response · Authors · 2026-05-12
> **Requested changes**
>
> ## Requested Changes
>
> * **1 Abstract:**
> * "we introduces..." change to "we introduce..."
> * "a Echo State..." change to "an Echo State..."
>
>
> * **2 Related Work:**
> * "parameters ,a" change to "parameters, a"
>
>
>
> **Author response:**
> We have corrected all typographical, grammatical, and spacing inconsistencies.
>
> * "we introduces" $\rightarrow$ "we introduce" in abstract,
> * "a Echo State" $\rightarrow$ "an Echo State" in abstract,
> * punctuation and spacing inconsistencies in the Related Work section "parameters ,a" $\rightarrow$ "parameters, a".
> * Additionally, we performed a complete proofreading pass across the manuscript to improve overall readability and presentation quality.
>
> * **Section 3.1 Problem Formulation:** What does $\mathcal{E}$ in the first line represent?
>
> **Author response:**
> In the revised manuscript, we explicitly define $\mathcal{E}$ as the edge set of the graph $\mathcal{G}=(\mathcal{V},\mathcal{E})$ in subsection 3.1 of methodology, where:
>
> * $\mathcal{V}$ denotes the set of traffic sensor nodes,
> * $\mathcal{E}$ represents the connectivity relationships between nodes.
>
> * **Section 3.3 Diffusion Convolution Implementation:**
> * What is $H$? Is it the same as $h$ from Section 3.1?
> * What is $P$? Is it an integer describing the matrix shape or the same $P$ from the previous section?
> * $H$ is also used for describing the matrix shape in $\mathbb{R}^{N \times (P+H)}$, which can be confusing.
>
>
>
> **Author response:**
>
> We thank the reviewer for identifying this ambiguity. We agree that there were several inconsistencies in math notations. In the revised manuscript we have removed the notation ambiguity and added a notations table in the appendix A for describing each of the symbols. We summarise the changes made for better understanding and notation consistency in the section 3 as follows:
>
> * $h$ in section 3.1 used earlier to define a predictive mapping is replaced by $\mathcal{F}$,
> * $H$ and $P$ used earlier in section 3.3 to represent the dimensional quantities in the following equation (1) are represented using separate notation ($H$ is replaced by $d_h$ and $P$ is replaced by $F$) as defined in notations table in appendix A,
>
> $$Z^{(t)} = [X^{(t)} \parallel H^{(t-1)}] \in \mathbb{R}^{N \times (F+d_h)}$$
>
>
> * $F$ represents the number of input features per node.
> * $d_h$ represents the dimension of the echo state network reservoir (hidden state).
> * $H^{(t-1)}$ in equation (1) represents the hidden state matrix at time step $(t-1)$.
> * Notation has been standardized throughout the manuscript to avoid symbol overloading.
> * $P$ is explicitly defined as stationary distribution of the diffusion process.
> * The definition is now provided immediately after its first appearance as well as in notations Table 6 in appendix A.
> * All symbols are consistently used throughout the manuscript.
>
> * **Section 3.3 continued:**
>
> 1. **Same thing: what are $P$ and $H$?**
> 2. **Add space after "spetial vector"**
> 3. **Start a new paragraph for the paragraph beginning with "3. Spatial Mixing".**
>
>
>
> **Author response:**
>
> We revised the manuscript to clearly define all variables immediately after their appearance. Specifically:
>
> 1. **Same thing: what are $P$ and $H$?**
>
> **Author response:**
>
> $H$ and $P$ used earlier in section 3.3 point 1 for Input Projection step to define dimensions of $W_{pre}$ are represented using separate notation ($H$ is replaced by $d_h$ and $P$ is replaced by $F$) as defined in notations Table 6 in appendix A.
> * $F$ represents the number of input features per node.
> * $d_h$ represents the dimension of the echo state network reservoir (hidden state).
>
>
> 2. **Add space after "spetial vector"**
>
> **Author response:**
> The spacing error has been corrected in the revised manuscript in point 2 of section 3.3.
>
> 3. **Start a new paragraph for the paragraph beginning with "3. Spatial Mixing".**
>
> **Author response:**
>
> "Spatial Mixing" paragraph has been reformatted into a separate paragraph in the revised manuscript in point 3 of section 3.3.

---

> ### Author Response · Authors · 2026-05-12
> **Requested changes**
>
> * **Section 3.4.1:**
> * You start by pointing out what $X$ is, but where is $X$ in Equation 5?
> * The last paragraph also uses $H$ for matrix size.
>
> **Author response:** We thank the reviewer for identifying this ambiguity. We agree that it was difficult to see the relation of input feature matrix $X$ with Equation (5). In the revised manuscript:
>
> * The role of the input feature matrix $X$ in Equation (5) has been clarified in the definition of $S^{(t)}$ i.e. spatial component in section 3.4.1. We explicitly connected it to the corresponding equation components. $S^{(t)}$ captures the multi-scale spatial dependencies of the current input $X^{(t)}$ and previous state $H^{(t-1)}$ via the Projection-Concatenation-Projection architecture described in subsection 3.3.
> * The surrounding explanation was revised for better consistency between notation and equations.
>
> **The last paragraph also uses $H$ for matrix size.**
> **Author response:**
> In the revised manuscript:
> * The overloaded use of $H$ for matrix dimensions was removed and $H$ was replaced by $d_h$ as defined in notations table to represent the matrix size.
> * Separate dimensional notation is now used consistently throughout the manuscript.
> * Notation consistency was improved across all mathematical formulations.
> * **Sections 5.1 and 5.2:**
> The sentence "To validate the ..." appears both at the end of Section 5.1 and the beginning of Section 5.2.
>
> **Author response:**
> We thank the reviewer for noting this redundancy. The repeated sentence has been removed from section 5.1 and retained in section 5.2 in the revised manuscript to improve conciseness and the overall flow of the experimental sections.
> * **Section 5.4: The phrase "robust near-zero generalization gap" is unclear. Please reword this sentence.**
>
> **Author response:**
> We thank the reviewer for highlighting the unclear phrasing. In the revised manuscript, we replaced this sentence with clearer wording explaining that DC-ESN maintains consistent performance between training and testing phases with minimal overfitting, indicating stable generalization behavior across datasets.
> * **Figure 1: Can you make this figure easier to read in black and white? Maybe use different line types as in Figure 7.**
>
> **Author response:**
> We thank the reviewer for this valuable suggestion. Figure 1 on page 2 has been redesigned to improve readability under grayscale and black-and-white printing conditions. We now use distinct line styles, marker patterns, and improved contrast so that all components remain distinguishable without relying solely on color.
> * **Tables 2 and 3: these seem to contain almost exactly the same information (except you add s.d. for different seed for DC-ESN in table 3), is there a way to simply combine these tables?**
> **Author response:**
> We agree with the reviewer that the presentation could be made more concise. Accordingly, we merged Tables 2 and 3 into a single unified table on page 14 in the revised manuscript. The updated table now reports both the primary evaluation metrics (in blue colour) and the corresponding standard deviations across random seeds (in red colour), improving clarity while reducing redundancy.
> * **Table 5: In the last row I see 1.706 as the "Best" and then "1.660" as the final. I'm a bit confused. Why is 1.660 not the best?**
>
> **Author response:**
> We thank the reviewer for carefully identifying this inconsistency. The issue resulted from a reporting/formatting error in the original table. We have corrected Table 5 in the revised manuscript and clarified the distinction between the best validation performance and the final reported test performance. The best validation MAE as per results is 1.706 and the final validation MAE is 1.707.
>
>
> We sincerely thank Reviewer again for the constructive feedback and positive assessment of our work. We have carefully addressed all comments and believe the revised manuscript is now substantially improved in terms of clarity, notation consistency, presentation quality, and readability.

---

### Review · Reviewer_v8WL · 2026-04-20

**Summary Of Contributions:**

In this paper, the authors study the problem of traffic time-series forecasting. To this end, they take DC-RNN model (DIFFUSION CONVOLUTIONAL RECURRENT NEURAL NETWORK: DATA-DRIVEN TRAFFIC FORECASTING, ICLR, 2018) and replace the RNN component with Echo State Networks (ESN). They compare their approach with simplistic baselines and DC-RNN and another model from 2018.

**Audience:**

No

**Audience Explanation:**

The paper's approach is outdated, with similar approaches being widely explored for time-series forecasting. Moreover, the compared methods are outdated (latest from 2018). The literature has advanced significantly since then.

**Broader Impact Concerns:**

Not necessary.

**Claims And Evidence:**

Yes

**Claims Explanation:**

The paper claims improvements in terms of the number of trainable parameters, the memory, and the inference speed. The experiments do support these claims.

**Requested Changes:**

Weaknesses:

1. The paper's novelty is very limited.

1.1. The literature has already extensively explored the combination of ESNs with Graph Neural Networks (GNNs) for various spatiotemporal modeling problems, including time-series forecasting and traffic time-series forecasting. With a quick search, one could easily find the following:

Han, X., & Zhao, Y. (2022). Interpretable graph reservoir computing with the temporal pattern attention. IEEE Transactions on Neural Networks and Learning Systems, 35(7), 9198-9212.

Zhang, L., & Zhang, G. (2025, June). ESSTPNet: Enhancing Spatio-Temporal Forecasting with Deep Echo State Networks and Adaptive Multi-Scale Decoding. In 2025 International Joint Conference on Neural Networks (IJCNN) (pp. 1-8). IEEE.

Sethi, S. (2024). Applying Reservoir Computing for Driver Behavior Analysis and Traﬀic Flow Prediction in Intelligent Transportation Systems (Doctoral dissertation, Virginia Tech).

Soussia, M., Salsabila, G. A., Mahjoub, M. A., & Rekik, I. (2026). Reservoir-Based Graph Convolutional Networks. IEEE Transactions on Pattern Analysis and Machine Intelligence.

1.2. The paper's contributions with respect to these very related studies addressing very similar issues on the same problem are unclear.

1.3. Section 2 should be more comprehensive, with dedicated sections on spatiotemporal ML models, time-series forecasting and traffic time-series forcasting, summarizing also key differences with respect to prior work.

1.4. "Existing deep learning approaches such as Convolutional Neural Networks (CNNs) for spatial modeling, Recurrent Neural Networks (RNNs), Long Short-Term Memory (LSTM), and Gated Recurrent Unit (GRU) networks for temporal modeling, as well as Graph Convolutional Networks (GCNs) for capturing topological structures have achieved notable success. But they often incur high computational cost and require large number of trainable parameters." => How about Mamba-based approaches?

2. Experimental evaluation is not convincing. The paper compares the proposed approach to very simplistic baselines (HA, ARIMA, VAR) or methods from 2018. The literature has progressed very significantly since then.


Minor comments:
- "we introduces" => "we introduce".
- "a Echo State Network" => "an Echo State Network".
- Introduction first paragraph: Providing references to e.g. review papers would be helpful here.
- "figure 1." => "Figure 1."
- "Recent advancements in deep learning, such as Graph Neural Networks (GNNs), have improved performance in non-Euclidean domains" => These are not recent any more.
- "Recent developments in Echo State Networks (ESNs) (Gallicchio et al., 2018; Gallicchio & Micheli, 2017)," => These are not recent.
- Fig 2: Text too small. Please use SVG or PDF.
- Fig 3: Low-quality bitmap.
- "Figure 3: The DC-ESN Cell architecture" => This is not the architecture but a single layer.
- Fig 4: It would be better to see in this figure the diffusion block as well.

---

> ### Author Response · Authors · 2026-05-12
> **Revised Manuscript Link**
>
> We sincerely thank the reviewers for their valuable feedback. We have uploaded our revised manuscript, which addresses their comments, the following link to view the revision: https://openreview.net/forum?id=9AOUXLYlwy&noteId=9AOUXLYlwy

---

> ### Author Response · Authors · 2026-05-12
> **Summary and Strengths**
>
> We sincerely thank Reviewer for the critical and constructive feedback. We thank for the acknowledgment that our claims regarding computational efficiency—specifically parameters, memory, and inference speed—are well supported by the experimental evidence. We have carefully addressed the concerns regarding novelty, related work, and experimental evaluation. Our detailed responses and the corresponding revisions are provided below:
>
> ## Summary and Strengths
>
> **Reviewer Comment:**
> *The paper claims improvements in terms of the number of trainable parameters, the memory, and the inference speed. The experiments do support these claims.*
>
> **Author response:**
> We thank the reviewer for validating the core empirical claims of our study. The primary objective of the DC-ESN is to provide a high-efficiency alternative for spatio-temporal forecasting, and we are pleased that the reviewer found the evidence for these efficiency gains to be accurate and convincing.
>
> ---

---

> ### Author Response · Authors · 2026-05-12
> **Requested changes**
>
> ## 1. Novelty and Positioning (Weaknesses 1.1 & 1.2)
>
> **Reviewer Comment:** *The paper's novelty is very limited. The literature has already extensively explored the combination of ESNs with Graph Neural Networks... The paper's contributions with respect to these studies are unclear.*
>
> **Author response:**
> We acknowledge the importance of positioning our work within the evolving landscape of Graph Reservoir Computing. We respectfully clarify that our contribution is not a direct substitution of GRU with ESN, but a structural reformulation of diffusion-based spatiotemporal modeling. Although the novelty is limited, the proposed framework is a simple and light-weight model, based on theory and intuition by Tobler's law for static graph construction. In many practical deployments, inference latency, memory footprint and hardware constraints are very critical. This motivates the central question of the proposed work i.e. Can competitive spatiotemporal traffic forecasting be achieved using substantially simpler recurrent dynamics than gated RNNs and modern high complexity architectures?
>
> Specifically, DC-ESN introduces a decoupled architecture where:
> * *(i)* spatial dependencies are modeled via trainable diffusion convolution which is theoretically much stronger than standard graph convolution, and
> * *(ii)* temporal dynamics are captured through a fixed reservoir with node-specific states.
>
> This differs from prior Graph Reservoir Computing approaches in two key ways:
>
> * We preserve the diffusion convolution mechanism of DCRNN, enabling modeling of directed traffic flow, which is not explicitly addressed in existing Graph RC frameworks.
> * We adopt a hybrid optimization strategy (which is different from backpropagation through time in full learnable architectures and readout only update based training strategy of GraphRC model) where gradients propagate through spatial components while temporal dynamics remain fixed, improving stability and efficiency. The following are elaborated fundamental architectural and functional differences between the proposed DC-ESN and the cited works:
>
> **1. Difference in Spatial Modeling and Reservoir design (vs. Han & Zhao):**
> Han and Zhao (2024) utilize a GraphRC framework that focuses on interpretability through the Kronecker product and Temporal Pattern Attention. It is a fully structural (not random) reservoir based model because it is designed and optimized using DAG (directed acyclic graph) with only output layer training. In contrast, DC-ESN introduces *Diffusion Convolution* into the reservoir cell and the reservoir used is simplest which is not designed using any specific procedure like "Improved GraphRC".  This is not merely a combination of GNN and ESN, but a physics-aware integration that models spatial dependencies as a stochastic diffusion process. This approach is mathematically more suited for directed transportation networks than the standard graph convolutions used in GraphRC. Since DC-ESN integrates Diffusion Convolution, this allows the model to capture the directional and stochastic nature of traffic flow (following the Tobler's law ), which is not explicitly addressed in the cited Reservoir-based GCNs.
>
> **2. Optimization for Efficiency:** The "Improved GraphRC"  relies on a structured Graph reservoir and a non-learnable attention mechanism with only readout training which make it computationally efficient. There is no doubt sometimes reservoir based models outperform deep learning (fully trainable) models, but they need structural design. As we can see from GraphRC paper, improved GraphRC outperforms all deep learning models on METR-LA data set which is a chaotic data set but it is small with only 207 nodes, but when its comes to PEMS-BAY data set with 325 sensors, the model accuracy for "Improved GraphRC" deprecates (looking at the results section of the paper "Improved GraphRC") despite the fact that PEMS data set is smoother and less noisy than METR-LA. Also as we are using simplest reservoir, so "Improved GraphRC" gives a scope that if we use or integrate a suitable structured reservoir, the performance of proposed DC-ESN can be improved significantly as we mention in future scope in the second point of "Future Directions" . DC-ESN prioritizes computational efficiency. By replacing the complex gated units of traditional models (like DCRNN) with fixed, sparse ESN cells, our framework achieves a **5.9-fold speedup** and **64% memory reduction**. Similar to "GraphRC", DC-ESN attains these efficiencies without the need for heavy trainable recurrent layers. "PCP" framework used for diffusion convolution learning makes the DCESN model efficient when spatial dependency matters as we have shown in results section on PEMS-BAY data set across various forecasting horizons and it shows comparable performance to most of the models with lower computational cost and low GPU usage which makes it more suitable for real-time applications on resource-constrained devices.

---

> ### Author Response · Authors · 2026-05-12
> **Requested change :**
>
> ## Novelty and positioning continued..
> **3. Complexity of ESSTPNet** : Although ESSTPNet (Zhang \& Zhang, 2025) also combines deep Echo State Networks with spatiotemporal forecasting, its architecture relies on a comparatively more complex adaptive multi-scale decoding framework optimized using reinforcement learning strategies. In contrast, the proposed DC-ESN adopts a significantly simpler and more computationally efficient design by directly building upon the established DCRNN encoder-decoder framework and replacing the fully trainable GRU-based temporal module with a fixed ESN reservoir. This design choice preserves the strong diffusion-based spatial modeling capability of DCRNN while substantially reducing training complexity, parameter count, GPU memory consumption, and optimization overhead without requiring additional reinforcement learning-based controller mechanisms or adaptive decoder search procedures.
>
> **4. Reproducibility constraint:**  We thank the reviewer for suggesting these additional ESN-GNN based studies. We carefully examined the proposed references and incorporated a detailed discussion of their methodologies and differences with respect to the proposed DC-ESN framework in the revised Related Work section. However, we were unable to include these approaches in the experimental comparison due to reproducibility limitations and the lack of publicly available implementations. In particular, several of the cited studies do not provide open-source code, complete hyperparameter configurations, or sufficient implementation details necessary for fair reproduction on the METR-LA and PEMS-BAY benchmarks. Furthermore, since our work explicitly emphasizes computational efficiency, memory reduction, and Green AI considerations, a meaningful comparison would additionally require consistent reporting of GPU memory consumption, training latency, parameter counts, and hardware configurations. These computational metrics were either not explicitly reported or not reproducible from the available descriptions in the cited works. To ensure fairness and experimental reliability, we therefore restricted our quantitative comparisons to methods with established benchmarks and reproducible implementations in the traffic forecasting literature.
>
> To address the reviewer’s concern regarding literature coverage, Related work section has been significantly expanded and reorganized. The revised section now includes dedicated discussions on:
> * statistical and classical traffic forecasting methods,
> * deep spatiotemporal learning approaches,
> * graph neural network-based traffic forecasting,
> * reservoir computing and ESN-based spatiotemporal forecasting
> * cited and discussed the suggested references (*Han & Zhao, 2022; Zhang & Zhang, 2025; Sethi, 2024; Soussia et al., 2026*).
> * Provided a more comprehensive comparison highlighting differences in graph modeling, temporal dynamics, and training strategy between prior graph reservoir methods and the proposed DC-ESN framework.

---

> ### Author Response · Authors · 2026-05-12
> **Requested changes:**
>
> * **2. Comprehensiveness of Literature (Weaknesses 1.3 & 1.4)**
> **Reviewer Comment:** *1.3. Section 2 should be more comprehensive, with dedicated sections on spatiotemporal ML models, time-series forecasting and traffic time-series forcasting, summarizing also key differences with respect to prior work. How about Mamba-based approaches?*
>
> **Author response:**
>
> We agree that the related work section should better reflect recent developments. In this context, we have significantly expanded the **Related Work (Section 2)** to bridge the gap between classical methods and modern architectures:
> * Added dedicated subsections for **Modern Spatio-Temporal GNNs**, **Attention-based Models**, **Mamba based and State space models** and **Reservoir computing based models** in the related section.
> * Included a discussion on selective state-space models and **Mamba-based approaches** for sequence modeling in the related work section.
> * We clarify that while high-capacity models like Mamba or Transformers achieve high accuracy, DC-ESN targets the *efficiency-critical* regime.
> * Summarized key differences of the proposed model with respect to prior work and key question addressed.
>
> * **3. Experimental Evaluation and Baselines (Weakness 2)**
>
> **Reviewer Comment:** *Experimental evaluation is not convincing. The paper compares the proposed approach to very simplistic baselines (HA, ARIMA, VAR) or methods from 2018. The literature has progressed very significantly since then.*
>
> **Author response:**
>
> We acknowledge the concern regarding the use of older baselines. In the revised manuscript, we have additionally incorporated newer and more recent baselines, including PM-MemNet and STAEformer, by reproducing and evaluating these models under consistent experimental settings. These additions strengthen the empirical evaluation and improve the positioning of the proposed DC-ESN framework with respect to recent advances in spatiotemporal forecasting. Earlier experiments primarily focused on classical and widely adopted baselines such as DCRNN to ensure reproducibility, stable benchmarking protocols, and fair comparison across models. However, we agree with the reviewer that comparisons against newer architectures are important for a comprehensive evaluation, and the revised manuscript now reflects this broader experimental analysis.
> In the revised manuscript we have done the following changes:
> * Added a discussion comparing DC-ESN with modern architectures (e.g., Transformer-based and Mamba-based models) in terms of computational complexity, parameter count, and scalability.
> * Included a comparative table highlighting the accuracy–efficiency trade-off across representative models.
> * Emphasized that DC-ESN is designed for efficiency-critical scenarios, where resource constraints are as important as predictive accuracy. We have added a performance-efficiency trade-off table that highlights how DC-ESN maintains competitive accuracy while achieving a **5.9-fold speedup** and **64% memory reduction** compared to the benchmark DCRNN.
>
> Importantly, we clarify that our work targets a complementary research direction: improving computational efficiency while maintaining competitive performance, rather than maximizing predictive accuracy alone.
>
> **4. Clarification of Contribution Scope**
>
> We would like to emphasize that the goal of DC-ESN is not to outperform the latest high-capacity models in absolute accuracy, but to provide a computationally efficient alternative that achieves competitive performance with significantly reduced training time and memory usage.
>
>  We would like to emphasize that the goal of DC-ESN is not to outperform the latest high-capacity models in absolute accuracy, but to provide a computationally efficient alternative that achieves competitive performance with significantly reduced training time and memory usage.
>
> This positions our work within the underexplored regime of efficiency-aware spatiotemporal modeling, which is particularly relevant for real-time and large-scale deployment.

---

> ### Author Response · Authors · 2026-05-12
> **Requested changes:**
>
> * **5. Minor Comments and Figures**
>
> **Reviewer Comment:**
> * Change "we introduces" to "we introduce".
> * Change "a Echo State Network" to "an Echo State Network".
> * Introduction first paragraph: Providing references to e.g. review papers would be helpful here.
> * Change "figure 1." to "Figure 1."
> * "Recent advancements in deep learning, such as Graph Neural Networks (GNNs), have improved performance in non-Euclidean domains" $\rightarrow$ These are not recent any more.
> * "Recent developments in Echo State Networks (ESNs) (Gallicchio et al., 2018; Gallicchio & Micheli, 2017)," $\rightarrow$ These are not recent.
> * **Fig 2:** Text too small. Please use SVG or PDF.
> * **Fig 3:** Low-quality bitmap.
> * Figure 3: The DC-ESN Cell architecture" $\rightarrow$ This is not the architecture but a single layer..
> * **Fig 4:** It would be better to see in this figure the diffusion block as well.
>
>
> **Author response:**
> We thank the reviewer for the detailed observations.
> * **Grammar:** All typographical errors (e.g., "we introduce", "an Echo State", "Figure 1") have been corrected in the abstract.
> * Provided references in the first paragraph of the introduction section.
> * **Phasing:** Removed the term "recent" for citations from 2017--2018.
> * **Visuals:** **Figure 2** has been updated using PDF for clarity. **Figure 3** has been redrawn at higher resolution and renamed to "The DC-ESN Cell layer" to reflect its function as a fundamental building block. **Figure 4** now explicitly illustrates the diffusion block.
>
>
> ## Conclusion
>
> We thank the Reviewer  for pushing us to better define the novelty of DC-ESN. By revising our literature survey to include modern Mamba and GraphRC frameworks and improving our visual documentation, we believe the manuscript now clearly demonstrates the value of the DC-ESN as a specialized, efficient solution for modern traffic forecasting.

---

### Review · Reviewer_NHar · 2026-04-28

**Summary Of Contributions:**

This paper proposes the Diffusion-Convolutional Echo State Network (DC-ESN) for traffic forecasting. Building on the existing Diffusion Convolutional Recurrent Neural Network (DCRNN), the paper proposes replacing the RNN component (in fact, GRU) with an Echo State Network (ESN). On the METR-LA and PEMS-BAY datasets, DC-ESN achieves almost the same accuracy as DCRNN, while improving inference speed and reducing GPU memory usage.

The paper is easy to understand. It is written with sufficient care so that even readers unfamiliar with research on traffic forecasting can understand the proposed method. The explanation of the proposed method is also appropriate, and the mathematical notation is carefully and clearly explained.

**Additional Comments:**

+ In the abstract, DC-ESN is defined twice.
+ P5: there is no space between "spatial vector" and $U^{(t)}$

**Audience:**

Yes

**Audience Explanation:**

Traffic forecasting is widely recognized as an application of machine learning, so there is no doubt that the paper falls within the scope of this journal. In addition, I believe that some readers would be interested in replacing the RNN component of DCRNN with an ESN, and thus that “at least some individuals in TMLR’s audience would be interested.” However, as discussed below, the impact of this work is not particularly large.

**Claims And Evidence:**

Yes

**Claims Explanation:**

At least within the scope of what is described in the paper, I believe that “the claims made in the submission are supported by accurate, convincing, and clear evidence.” However, I still have reservations about whether there is a compelling necessity to pursue those claims; please see the Requested Changes below.

**Requested Changes:**

The paper does not compare the proposed method with approaches that use attention mechanisms or Transformers for traffic forecasting. These methods are not discussed in the Related Work section in Section 2. I feel that the literature cited by the current manuscript is somewhat outdated. Nor are these methods included as baselines in the experiments in Section 5. Please cite the recent work (for example, the papers listed below; they are not a comprehensive list), position the proposed method and the paper’s claims in relation to them, and compare the experimental results against these methods. Based on recent prior work, please clarify the limitations of existing studies, the research question or challenge addressed by this work, and the impact of the present study.

+ Walid Guettala, Yufan Zhao, László Gulyás. 2025. Less is More: Strategic Expert Selection Outperforms Ensemble Complexity in Traffic Forecasting. IEEE ICTAI.
+ Renhe Jiang, Zhaonan Wang, Jiawei Yong, Puneet Jeph, Quanjun Chen, Yasumasa Kobayashi, Xuan Song, Shintaro Fukushima, Toyotaro Suzumura. 2023. MegaCRN: Meta-Graph Convolutional Recurrent Network. AAAI.
+ Hyunwook Lee, Seungmin Jin, Hyeshin Chu, Hongkyu Lim, Sungahn Ko. 2022. Learning to Remember Patterns: Pattern Matching Memory Networks for Traffic Forecasting. ICLR.

---

> ### Author Response · Authors · 2026-05-12
> **Revised Manuscript Link**
>
> We sincerely thank the reviewers for their valuable feedback. We have uploaded our revised manuscript, which addresses their comments, the following link to view the revision: https://openreview.net/forum?id=9AOUXLYlwy&noteId=9AOUXLYlwy

---

> ### Author Response · Authors · 2026-05-12
> **Requested Changes**
>
> We would like to thank Reviewer for the constructive feedback and for
> acknowledging the clarity of our mathematical notation and methodology. We
> sincerely thank the reviewer for insight regarding the positioning of our work relative to
> modern attention-based models and Transformers and agree with the suggested
> changes. In response to these suggestions, we have significantly extended the Related
> Work (Section 2), added a Literature Comparison Table, and included
> additional baseline comparisons (Appendix B). We summarize the specific changes made in revised manuscript as follows:
>
>
> **Summary and strengths :**
>
> **Reviewer Comment:**  *This paper proposes the Diffusion-Convolutional Echo State Network (DC-ESN) for traffic forecasting. The paper is easy to understand. It is written with sufficient care so that even readers unfamiliar with research on traffic forecasting can understand the proposed method. The explanation of the proposed method is also appropriate, and the mathematical notation is carefully and clearly explained.*
>
> **Author response:**
> We thank the reviewer for the positive assessment of our manuscript's clarity and for recognizing the effectiveness of DC-ESN in achieving competitive accuracy while improving inference speed and reducing GPU memory usage. We are encouraged that the mathematical notation was found to be clear and accessible to a broad audience.
>
> ## Requested Changes
>
> * **1. Comparison with Transformers and Attention Mechanisms**
>
> **Reviewer Comment:** *The paper does not compare the proposed method with approaches that use attention mechanisms or Transformers for traffic forecasting... I feel that the literature cited by the current manuscript is somewhat outdated. Please cite the recent work (for example, the papers listed below; they are not a comprehensive list), position the proposed method and the paper’s claims in relation to them, and compare the experimental results against these methods.*
>
> **Author response:**
> We acknowledge that the initial manuscript focused primarily on Graph Convolutional Recurrent Networks. In the revised manuscript:
> * We have updated the **Related Work (Section 2)** to include a comprehensive discussion of Transformer-based models and recent spatial-temporal attention frameworks on page numbers 3, 4, and 5 of the revised manuscript.
> * We added a **Literature Comparison Table 7** in the Appendix B that positions DC-ESN against these high-capacity models. For the models whose results were reproducible, we compared it with DC-ESN and added in the experiments section.
> * While Transformers offer high accuracy, they often suffer from quadratic computational complexity. We clarify that DC-ESN serves as a high-efficiency alternative, providing GCN-level spatial awareness with the minimal training overhead characteristic of ESNs.
>
>
>
> * **2. Addressing Recent Literature (2022–2025)**
>
> **Reviewer Comment:** *Based on recent prior work (e.g., MegaCRN, Pattern Matching Memory Networks), please clarify the limitations of existing studies, the research question or challenge addressed by this work, and the impact of the present study.*
>
>
> **Author response:**
> We have cited and integrated the suggested papers (*Jiang et al., 2023; Lee et al., 2022; Guettala et al., 2025*) into our related work section.
>
> -- **Positioning:** We explicitly state that while models like **MegaCRN** achieve state-of-the-art results through meta-learning, they require significant computational resources for gradient-based training.
>
> -- **Impact and Research Question being addressed:** The central question addressed by our study is "Can competitive spatiotemporal traffic forecasting be achieved using substantially simpler recurrent dynamics than gated RNNs and modern high-complexity architectures?" By utilizing an ESN, we eliminate the need for Backpropagation Through Time (BPTT) in the recurrent layer, resulting in a model that is significantly faster to train than memory-augmented or meta-learning networks, with minimal loss in accuracy.
>
>
> * **3. Additional comments: Minor Corrections (Abstract and P5)**
> **Reviewer Comment:**
>
> * *In the abstract, DC-ESN is defined twice.*
> * *P5: space between "spatial vector" and $U^t$*
>
>
> **Author response:**
> We thank the reviewer for their careful reading.
> * We have removed the redundant definition of DC-ESN from the **Abstract**.
> * The spacing error following "spatial vector" on **Page 6** in the revised manuscript has been corrected.
> * We have performed an additional proofreading pass to ensure no other typographical errors remain.
>
>
> We sincerely thank Reviewer again for the constructive feedback. We believe the inclusion of recent Transformer-based baselines and the clarified positioning of DC-ESN relative to state-of-the-art meta-learning approaches has significantly strengthened the impact and context of our study.

---

### Decision · Action_Editor_ohpB · 2026-06-05

**Recommendation:** Reject

**Audience:**

No

**Audience Explanation:**

While TMLR's audience is broadly interested in the problem of efficient spatiotemporal traffic forecasting, the specific findings of this paper are unlikely to hold their interest, because the contributions are framed and evaluated almost entirely as an improvement over DCRNN, which is a 2018 model that is no longer the state of the art. A reader evaluating whether to adopt or build on this work cannot conclude anything about its standing relative to the methods that define the field now, and this has been the primary concerns of the reviewers. The paper is not asking the right research question.

**Claims And Evidence:**

Yes

**Claims Explanation:**

The claims are supported by clear and convincing evidence. The efficiency gains (≈4.9–5.9× faster training, fewer parameters, lower GPU memory) and the near-parity accuracy are demonstrated consistently across both METR-LA and PEMS-BAY.

The weakness is not in evidentiary support but in the significance of the benchmark. The submission frames its contribution almost entirely as efficiency-with-parity against DCRNN, a 2018 model. By the paper's own tables, DCRNN is no longer the accuracy frontier—STAEformer clearly outperform it. So while the claims are well-backed, matching an outdated baseline limits the contribution's relevance, and the paper does not show comparable efficiency advantages over the more recent high-capacity models it concedes are stronger.

In addition, the reviewers suggested several other recent baselines to be incorporated. While the related work in the revision is significantly expanded, not all of them were considered as baselines.